# Finding the Correct Visual Evidence Without Forgetting: Mitigating Hallucination in LVLMs via Inter-Layer Visual Attention Discrepancy

Yutong Xie [1]   Zhenglin Hua [1]   Ran Wang [2 3]   Wing W. Y. Ng [4]   Xizhao Wang [5]   Yuheng Jia [1 6]

## Abstract

Large Vision-Language Models (LVLMs) have shown remarkable performance on a wide range of vision-language tasks. Despite this progress, they are still prone to hallucination, generating responses that are inconsistent with visual content. In this work, we find that LVLMs tend to hallucinate when they pay insufficient attention to the correct visual evidence and gradually forget it during the generation process. We empirically find that although LVLMs overall attend insufficiently to visual evidence, they exhibit sensitivity to the correct visual evidence in specific layers, with notable inter-layer discrepancy. Motivated by this observation, we propose a novel hallucination mitigation method that enhances visual evidence based on **I**nter-**L**ayer **V**isual **A**ttention **D**iscrepancy (**ILVAD**). Specifically, we obtain the attention weights from early generated tokens to visual tokens across layers and identify the tokens that are repeatedly activated as visual evidence, forming a saliency map. We then enhance attention to visual evidence during generation through the saliency map to reduce visual forgetting. In addition, we leverage the saliency map to obtain attention scores of generated text to visual evidence, in order to select and emphasize text tokens that are strongly grounded in visual evidence. Our method is training-free and plug-and-play. Multiple bench-mark evaluations conducted on five recently released models show that our method can consistently mitigate hallucinations in different LVLMs over various architectures. Code is available at https://github.com/ytx-ML/ILVAD.

## 1. Introduction

Large Vision-Language Models (LVLMs) (Liu et al., 2023; Zhu et al., 2024; Wang et al., 2024; Bai et al., 2025; Zhu et al., 2025) have achieved powerful cross-modal capabilities by aligning visual information with Large Language Models (LLMs), making significant progress in various vision-language tasks such as image captioning, visual question answering, grounded reasoning, and so on (Wang et al., 2023; Chen et al., 2025; Peng et al., 2024; Wang et al., 2026). However, LVLMs are still prone to hallucination, generating responses that are inconsistent with visual content. This issue seriously affects the reliability of LVLMs in high-risk areas such as medical diagnosis (Huang et al., 2025) and autonomous driving (Zhang et al., 2025a).

To mitigate hallucination, several studies introduce external knowledge bases for alignment training or fine-tuning (Zhao et al., 2023; Yu et al., 2024), but they incur significant computational and storage overhead. In comparison, recent studies have focused on train-free approaches during the inference stage. Among these, the decoding strategies (Leng et al., 2024; Kim et al., 2024; An et al., 2025; Wan et al., 2025; Huo et al., 2025) adjust the logits distribution during generation to reduce the influence of language priors. However, these methods operate primarily on the logits rather than the internal visual attention, thus failing to effectively promote LVLMs to focus on visual information. Some studies analyze hallucinations from the perspective of attention patterns. They observe that LVLMs tend to allocate excessive attention to visual tokens unrelated to the current query, e.g., background regions, referred to as visual attention sinks (Kang et al., 2025; Tang et al., 2025; Zhang et al., 2025b). Although reallocating attention can partially mitigate hallucinations, it remains limited when language priors remain dominant, as it does not genuinely increase the model's reliance on visual information.

---

[1]School of Computer Science and Engineering, Southeast University, Nanjing, China [2]School of Artificial Intelligence, Shenzhen University, Shenzhen, China [3]National Engineering Laboratory for Big Data Systems Computing Technology, Shenzhen University, Shenzhen, China [4]School of Computer Science and Engineering, South China University of Technology, Guangzhou, China [5]College of Computer Science and Software Engineering, Shenzhen University, Shenzhen, China [6]Key Laboratory of New Generation Artificial Intelligence Technology and Its Interdisciplinary Applications (Southeast University), Ministry of Education, China. Correspondence to: Yuheng Jia <yhjia@seu.edu.cn>.

*Proceedings of the 43rd International Conference on Machine Learning*, Seoul, South Korea. PMLR 306, 2026. Copyright 2026 by the author(s).

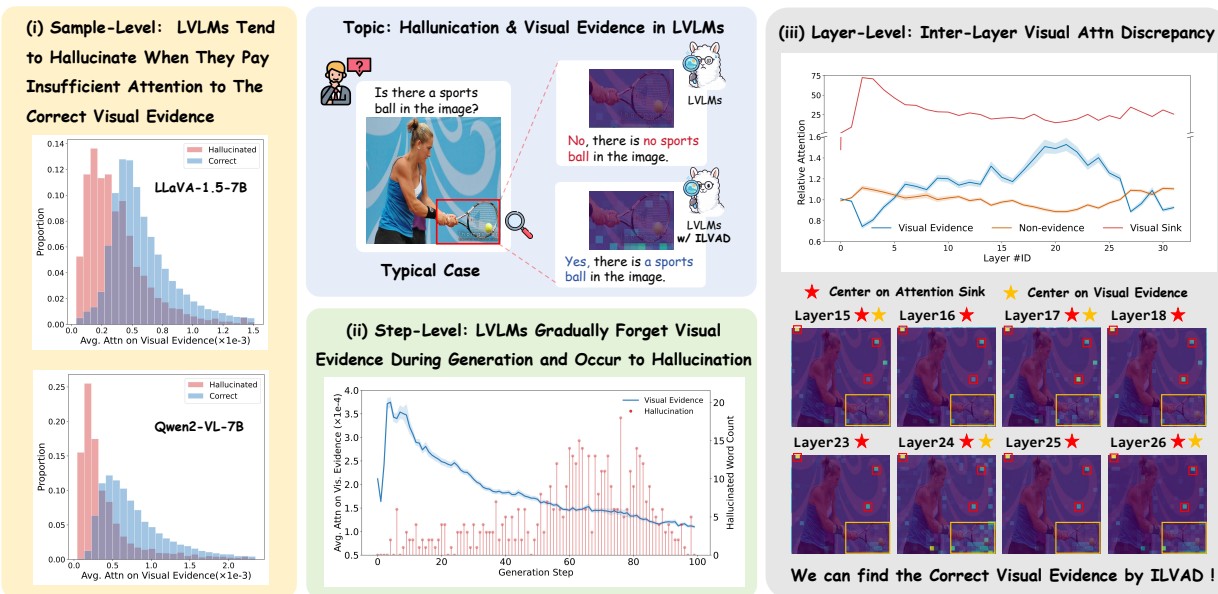

Figure 1. This figure showcases our insights. The (0) blue box illustrates that LVLMs are prone to hallucination due to the neglect of relevant visual evidence. The baseline model hardly attends to the visual evidence, while our method correctly identifies the object by enhancing attention to the visual evidence. The (i) yellow box shows the attention distribution over visual evidence for hallucinated versus correct textual responses at the sample-level in LLaVA-1.5-7B and Qwen2-VL-7B. The (ii) green box illustrates how the model's attention to visual evidence and the frequency of hallucinations change over the generation step-levels. The (iii) gray box shows the attention to visual evidence, visual sinks, and non-evidence across different layers. Attention exhibits Inter-Layer Visual Attention Discrepancy at the layer-level. The example below more clearly demonstrates this observation through the attention distribution at specific layers.

A common limitation of the above methods is that they do not effectively guide the model to consistently focus on visual evidence, i.e., visual contents that are relevant to the current query and support correct answers. We observe that LVLMs are more prone to hallucination when the correct visual evidence is ignored during inference. The (0) blue box in Figure 1 illustrates an example that hallucinations can be attributed to the neglect of relevant visual evidence. Although the correct visual evidence is clearly present, the model attends to irrelevant regions and generates responses inconsistent with the image.

To further illustrate the relationship between visual evidence and hallucinations, we conduct a quantitative analysis of visual attention patterns. The (i) yellow box in Figure 1 illustrates the average attention assigned to visual evidence between correct and hallucinated responses. The results reveal that hallucinated samples are predominantly associated with lower attention to visual evidence, while correct responses tend to allocate higher attention to them. This observation suggests that LVLMs tend to hallucinate when they pay insufficient attention to the correct visual evidence. Furthermore, we analyze attention dynamics over the generation steps. As shown in the (ii) green box of Figure 1, the attention to visual evidence gradually decreases, while the frequency of hallucinated content increases. This observation suggests that LVLMs tend to forget critical visual evidence during long-form generation, which further leads

to more hallucinations. These results confirm the strong correlation between attention to visual evidence and hallucinations. Thus, it is essential to promote the model to identify the correct visual evidence and consistently enhance its attention on it during generation.

To find the correct visual evidence, we analyze the model's attention patterns at different layers. The upper half of the (iii) gray box in Figure 1 shows the relative attention of visual evidence, visual sinks, and non-evidence across layers, i.e., the ratio to the layer-wise average attention. It is obvious that LVLMs focus on relevant visual evidence at specific layers, where the attention to visual evidence is above average and more distinguishable from non-evidence. However, selecting tokens based on relative attention does not filter out attention sinks since the model also assigns excessive attention to attention sinks. An overlooked observation is that the model's attention to visual evidence exhibits significant variation across layers, i.e., **I**nter-**L**ayer **V**isual **A**ttention **D**iscrepancy (**ILVAD**). As shown in the lower half of the (iii) gray box in Figure 1, while the LVLM shows attention sinks across most layers, it attends to visual evidence only in certain layers. Even in adjacent layers, there are significant differences in attention to visual evidence.

Motivated by this observation, we propose a method to mitigate hallucinations via ILVAD. Specifically, we first extract the average attention from early generated tokens to each

visual token across layers and leverage the inter-layer discrepancy to identify visual evidence and construct a saliency map. We then enhance attention to these evidence tokens through the saliency map to mitigate visual forgetting. Moreover, we leverage the saliency map to select and emphasize text tokens that are well-grounded in the visual evidence. Our method is training-free and plug-and-play. Multiple benchmark evaluations conducted on five recently released models show that ILVAD effectively mitigates hallucinations and improves visual comprehension in LVLMs across various model architectures.

Our contributions are summarized as follows:

- We analyze the relationship between visual evidence and hallucinations in LVLMs. Our analysis shows that hallucinations often occur when the model allocates insufficient attention to relevant visual evidence, and this issue worsens as the model gradually forgets this evidence during the generation process.

- We observe that LVLMs do not consistently attend to visual evidence across layers. Instead, attention to evidence shows strong inter-layer discrepancy. Motivated by this observation, we propose a train-free method that uses ILVAD to construct a visual evidence saliency map and enhance grounded generation.

- We evaluate our method on hallucination benchmarks and comprehensive benchmarks. Results show that our method effectively mitigates hallucinations and improves visual understanding in LVLMs.

## 2. Related Work

**Hallucination Mitigation in LVLMs.** Hallucination in Large Vision-Language Models (LVLMs) refers to the phenomenon in which the generated text is inconsistent with the actual visual content of the input image (Liu et al., 2024c; Hua et al., 2025). This issue severely affects the reliability of LVLMs in real-world applications. Existing studies on mitigating hallucinations in LVLMs can be broadly grouped into four categories: alignment methods during training (Yu et al., 2024), post-processing methods (Fang et al., 2025; Yin et al., 2024), decoding strategies (Leng et al., 2024; Kim et al., 2024; An et al., 2025), and attention-based analysis and intervention methods (Yin et al., 2025; Kang et al., 2025; Jung et al., 2025; Wan et al., 2025; He et al., 2025; Zhang et al., 2025b). The first two categories typically rely on additional external resources or auxiliary models, while the proposed ILVAD method operates purely on the model's internal states and does not require any external information.

**Contrastive Decoding in LVLMs.** Existing decoding strategies (Leng et al., 2024; Kim et al., 2024) mainly intervene at the output level by adjusting the logits distribution during

generation to reduce the influence of language priors. Recent adaptations of this technique combine decoding with attention: AGLA (An et al., 2025) masks irrelevant and performs collaborative decoding combining the output of the original image and the masked image. ONLY (Wan et al., 2025) defines text-to-visual attention entropy ratio to enhance decoding by selecting attention heads. However, these methods do not explicitly enhance the model's internal attention to visual information. In contrast, ILVAD constructs a visual evidence saliency map based on inter-layer attention differences and guides effective enhancement of visual information from within the model.

**Attention Intervention in LVLMs.** Another line of research analyzes hallucination through internal attention behaviors, showing its close link to visual token attention during generation. VAR (Kang et al., 2025) observes that LVLMs commonly exhibit the visual attention sink, where certain visual tokens consistently receive high attention across different queries. This method reduces hallucination by redistributing attention from sink tokens to others. EVAS (Zhang et al., 2025b) further analyzes the density of visual attention sinks in early layers and finds that a higher density of visual-centric heads is correlated with fewer hallucinations. Beyond attention sinks, recent work also focuses on dynamic attention recalibration during generation. SPARC (Jung et al., 2025) identifies key visual tokens using attention differences across time steps and progressively reinforces them. VHR (He et al., 2025) identifies attention heads that are highly sensitive to visual inputs and reinforces them to reduce language bias. Although these methods mitigate hallucinations to some extent, they share a common limitation: they do not effectively guide the model to maintain sustained attention to visual evidence relevant to the current query during generation. In contrast, ILVAD leverages inter-layer attention discrepancy to identify visual evidence and construct a saliency map and continuously enhances the attention to visual evidence tokens during generation, which effectively reduces hallucinations and improves visual understanding in LVLMs.

## 3. Method

In this section, we first introduce the notions of LVLMs. Then we detail the extraction of the saliency map via Inter-Layer Visual Attention Discrepancy. Finally, we enhance visual evidence attention to promote grounded generation. The overview of our ILVAD method is presented in Figure 2.

### 3.1. Preliminaries

**LVLM generation.** LVLMs typically consist of a vision encoder, a modality connector, and an autoregressive language model. Given a system query, a raw image and a textual prompt as input, the raw image is encoded and trans-

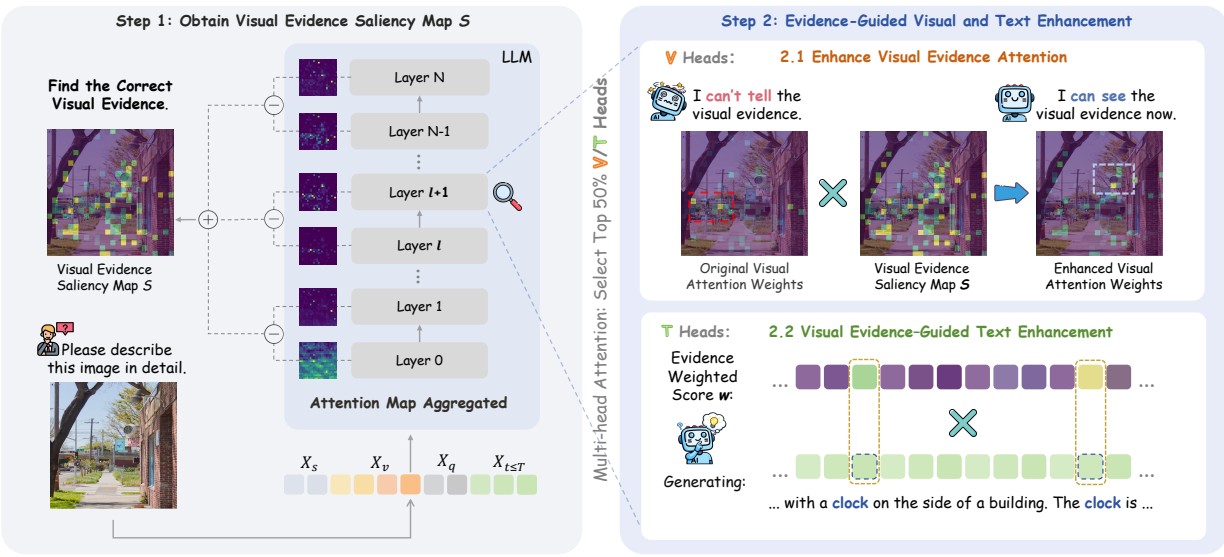

Figure 2. Overview of the proposed method Inter-Layer Visual Attention Discrepancy (ILVAD). We extract the visual evidence saliency map in the first stage. Given the attention weights from the generated text tokens in layers, we aggregate the attention by averaging over the first $T$ generation steps. Then we use the threshold to select the salient tokens in each layer to filter out irrelevant areas. Then we perform layer-by-layer subtraction to remove attention sinks. Finally, we accumulate the subtraction and normalize to obtain the visual evidence saliency map. In the second stage, we enhance attention to visual evidence during generation through the saliency map to reduce visual forgetting. Moreover, we leverage the saliency map to obtain attention scores of generated text to visual evidence, in order to select and emphasize text tokens that are strongly grounded in visual evidence. Our method is training-free and plug-and-play, which is applicable to various model architectures to mitigate hallucinations.

formed into visual tokens $\boldsymbol{X}_v$ through the vision encoder and the modality connector. Meanwhile, the system query and textual query are tokenized as system tokens $\boldsymbol{X}_s$ and query tokens $\boldsymbol{X}_q$, respectively. These components are concatenated as initial input $\boldsymbol{X} = [\boldsymbol{X}_s, \boldsymbol{X}_v, \boldsymbol{X}_q] \in \mathbb{R}^{n \times d}$, where $n$ and $d$ are the number and dimension of input tokens, respectively. At each time step $t$, the model predicts the next token conditioned on the input tokens and the previously generated tokens in an autoregressive manner:

$$y_t = \arg\max p_\theta(y_t | \boldsymbol{X}, \boldsymbol{y}_{<t}), \tag{1}$$

where $y_t$ and $\boldsymbol{y}_{<t}$ are the next predicted token and previously generated tokens, respectively.

**Attention Mechanism.** The language model is based on the Transformer architecture. Denote $\boldsymbol{X}^{l-1} = \{\boldsymbol{x}_1, \boldsymbol{x}_2 ..., \boldsymbol{x}_{n'}\}$ as the input of layer $l$. At the $l$-th layer and $h$-th attention head, the attention weights are given by:

$$\boldsymbol{A}^{l,h} = Softmax\left(\frac{\boldsymbol{Q}^{l,h}(\boldsymbol{K}^{l,h})^\top}{\sqrt{d_k}} + M\right), \tag{2}$$

where $\boldsymbol{Q}^{l,h} = \boldsymbol{X}^{l-1}\boldsymbol{W}_Q^{l,h}$ and $\boldsymbol{K}^{l,h} = \boldsymbol{X}^{l-1}\boldsymbol{W}_K^{l,h}$ represent the query and key matrices from the learned projection matrices, respectively. $Softmax(x) = e^{x_i} / \sum_i e^{x_i}$ is the activation function. $M$ represents the casual mask. $d_k$ is the scaling factor. The attention weight $A_{i,j}^{l,h}$ represents the

degree of attention that the $i$-th token pays to the $j$-th token. The multi-head attention (MHA) mechanism concatenates the outputs of all attention heads and projects them back to the original dimension:

$$MHA^l(\boldsymbol{X}^{l-1}) = [\boldsymbol{A}^{l,1}\boldsymbol{V}^{l,1}, ..., \boldsymbol{A}^{l,H}\boldsymbol{V}^{l,H}]\boldsymbol{W}_O^l, \tag{3}$$

where $\boldsymbol{V}^{l,h} = \boldsymbol{X}^{l-1}\boldsymbol{W}_V^{l,h}$ represents the value matrix from the learned projection matrix. $\boldsymbol{W}_O^{l,h}$ is the output projection matrix. $H$ is the number of attention heads in each layer.

### 3.2. Inter-Layer Visual Attention Discrepancy

Inspired by the findings in Section 1, we propose a method via inter-layer visual attention discrepancy to identify visual evidence. Given the attention weights $\boldsymbol{A}^l$ from the generated text tokens in layer $l$, we aggregate the attention by averaging over the first $T$ generation steps. Since different attention heads exhibit varying sensitivity to visual information (Kang et al., 2025), we only retain the top 50% of attention heads ranked by total visual attention sum. For each layer, the average attention weights over these heads is given by:

$$\bar{A}_j^l = \frac{1}{T \cdot |\boldsymbol{H}_v^l|} \sum_{h \in \boldsymbol{H}_v^l} \sum_{i \in \boldsymbol{X}_{t \leq T}} A_{i,j}^{l,h}, j \in \boldsymbol{X}_v, \tag{4}$$

where $\boldsymbol{X}_{t \leq T}$ and $\boldsymbol{X}_v$ represent the set of the first $T$ generated text tokens and visual tokens, respectively. $\boldsymbol{H}_v^l$ represents the set of visually sensitive attention heads of the layer $l$.

---

**Algorithm 1** Pseudo-code of ILVAD

---

**Require:** image $I$, query $q$, LVLM $\mathcal{M}$ with $H$ heads and $L$ layers; first-token window $T$; threshold $\tau$; strengths $\alpha, \beta$; head ratio $\rho = 0.5$.
**Ensure:** enhanced attention weights $\hat{A}$.

1: Run $\mathcal{M}(I, q)$ to obtain attentions from the first $T$ generated tokens: $A^{l,h}$;
2: $H_v^l \leftarrow \text{SELECTVISHEADS}(A^{l,h}, \rho)$;
3: **for** $l = 1$ to $L$ **do**
4:     $\bar{A}^l \leftarrow \text{AVGVISATTN}(A^{l,h \in H_v^l}, T)$;      ▷ Eq.(4)
5:     $\tilde{A}^l \leftarrow \text{BINARIZE}(\bar{A}^l, \tau)$;          ▷ Eq.(5)
6: **end for**
7: **for** $l = 1$ to $L - 1$ **do**
8:     $S \leftarrow S + \max(\tilde{A}^{l+1} - \tilde{A}^l, 0)$;     ▷ Eq.(6)
9: **end for**
10: **for** $l = 1$ to $L$ **do**
11:     $\hat{S} \leftarrow \text{NORMALIZE}(S)$;
12:     $e^{l,h} \leftarrow \text{EVIRATIO}(A^{l,h}, \hat{S})$;      ▷ Eq.(7)
13:     $H_e^l \leftarrow \text{SELECTEVIHEADS}(e^{l,h}, \rho)$;
14:     $\hat{A}^{l,h} \leftarrow \text{ENHVISATTN}(A, \hat{S}, H_e^l, \alpha)$;    ▷ Eq.(8)
15:     $w \leftarrow \text{EVIWEIGHT}(\hat{A}^{l,h}, \hat{S}, \rho)$;     ▷ Eq.(9)
16:     $\hat{w} \leftarrow \text{NORMALIZE}(w)$;
17:     $H_t^l \leftarrow \text{SELECTTEXTHEADS}(A^{l,h}, \rho)$;
18:     $\hat{A}^{l,h} \leftarrow \text{ENHTEXTATTN}(\hat{A}^{l,h}, \hat{w}, H_t^l, \beta)$;   ▷ Eq.(10)
19: **end for**
20: $\hat{A} \leftarrow \text{NORMALIZE}(\hat{A})$;
21: **return** $\hat{A}$.

---

We then identify salient visual tokens at each layer by thresholding. A visual token is considered salient if its attention value exceeds a fixed multiple of the layer-wise mean attention:

$$\tilde{A}_j^l = \begin{cases} 1, & \text{if } \bar{A}_j^l > \tau \cdot \text{mean}(\bar{A}^l), \\ 0, & \text{otherwise,} \end{cases} \quad (5)$$

where $\text{mean}(\cdot)$ represents the average value over all visual tokens. $\tau$ is the threshold. This operation filters out non-evidence tokens that the model does not focus on and retain the visual evidence in certain layers. However, visual attention sinks are also recognized as salient tokens in nearly every layer, since the model allocates higher attention to them. To distinguish the correct visual evidence tokens, we define a visual token as activated only when it is newly identified as a salient token in a layer but not in the preceding one. The cumulative activation map is then defined as:

$$S = \sum_{l=1}^{L-1} \max(\tilde{A}^{l+1} - \tilde{A}^l, 0), \quad (6)$$

where $S_j$ represents the number of times that the $j$-th visual token is activated. This operation filters out visual attention

sinks. Finally, we normalize $S$ to obtain the saliency map of visual evidence $\hat{S} \in [0, 1]^{|X_v|}$ for subsequent enhancement. Figure 2 shows the saliency map from LLaVA-1.5-7B. Compared to the original attention map, our method effectively highlights visual evidence relevant to the query. For more visual evidence saliency maps, please refer to Appendix E.1.

### 3.3. Evidence-guided Visual and Text Enhancement

Given the visual evidence saliency map $\hat{S}$ obtained in Section 3.2, we further enhance the attention to visual evidence to prevent it from being diluted or forgotten during generation. We also strengthen text tokens grounded in visual evidence to reduce reliance on linguistic priors.

We first select the attention heads that are more sensitive to visual evidence in layer $l$. For the $i$-th generated text token, we define the evidence ratio as:

$$e_i^{l,h} = \frac{\sum_{j \in X_v} \hat{S}_j \cdot A_{i,j}^{l,h}}{\sum_{j \in X_v} A_{i,j}^{l,h}}, i \in X_t, \quad (7)$$

where $e_i^{l,h}$ represents the sensitivity of the attention head $h$ in layer $l$ to visual evidence when generating the $i$-th token. We select the top 50% of the attention heads as $H_e^l$ to enhance in each layer. Referring to studies on attention steering (Zhang et al., 2024), we enhance visual evidence attention on these attention heads at each generation step:

$$\hat{A}_{i,j}^{l,h} = A_{i,j}^{l,h} \cdot \exp(\alpha \hat{S}_j), h \in H_e^l, i \in X_t, j \in X_v, \quad (8)$$

where $\alpha$ controls the enhancement strength.

For each generated text token, we measure its reliance on visual evidence and emphasize those grounded in salient regions by evidence weighted score:

$$w_i = \frac{1}{L \cdot |H_t^l|} \sum_{h \in H_t^l} \sum_{j \in X_v} \sum_{l=1}^{L} \hat{S}_j \cdot A_{i,j}^{l,h}, i \in X_t, \quad (9)$$

where $L$ is the total number of layers. We only retain the top 50% of attention heads ranked by total visual attention sum as $H_t^l$, which represents the set of text sensitive attention heads of the layer $l$. We then normalize $w$ as $\hat{w} \in [0, 1]^{|X_t|}$ to enhance text tokens grounded in visual evidence:

$$\hat{A}_{i,j}^{l,h} = A_{i,j}^{l,h} \cdot (\hat{w}_i + \beta), h \in H_t^l, i \in X_t, j \in X_{<t}, \quad (10)$$

where $\beta$ is a control parameter that regulates the degree to which text tokens are enhanced or suppressed based on their attention weights to visual evidence. $X_{<t}$ represents the set of text tokens generated before the current text token. Finally, we normalize the enhanced attention weights by $\hat{A}_{i,j}^{l,h} = \hat{A}_{i,j}^{l,h} / \sum_j \hat{A}_{i,j}^{l,h}$. The overall pseudo-code of our IL-VAD method is summarized in Algorithm 1. Our method

Table 1. Performance evaluation on hallucination benchmarks, where **bold** and underlined indicate the best and second best results respectively. We limit the maximum number of new tokens to 512 for the CHAIR benchmark. The MMHal-Bench is scored by GPT-4o.

| Model | Method | CHAIR | | | POPE | | MMHal-Bench | |
|---|---|---|---|---|---|---|---|---|
| | | CHAIR$_S$ ↓ | CHAIR$_I$ ↓ | Len. | Accuracy↑ | F1 Score↑ | Hal. ↓ | Score↑ |
| LLaVA-1.5-7B | Greedy | 48.6 | 13.52 | 82.66 | 84.50 | 85.22 | 67.0 | 2.01 |
| | Beam | 52.1 | 13.70 | 93.28 | 85.41 | 85.40 | 67.2 | 2.00 |
| | VCD | 48.4 | 13.47 | 85.93 | 84.74 | 85.31 | 61.8 | 2.20 |
| | CODE | 48.2 | 13.34 | 78.34 | 84.19 | 85.06 | 66.2 | 2.05 |
| | AGLA | 46.4 | 13.27 | 89.74 | 85.63 | **86.38** | 63.8 | 2.14 |
| | VAF | 49.8 | 13.28 | 89.38 | 83.03 | 84.43 | 68.2 | 1.97 |
| | VAR | 52.5 | 14.17 | 88.72 | 84.82 | 85.87 | 62.2 | 2.18 |
| | SPARC | 48.9 | 13.42 | 89.44 | 84.60 | 85.28 | 64.6 | 2.09 |
| | ONLY | 47.8 | 13.23 | 95.85 | 85.13 | 85.54 | 69.8 | 1.88 |
| | VHR | 34.5 | 9.86 | 81.73 | 84.74 | 85.45 | 65.6 | 2.10 |
| | **Ours** | **32.6** | **9.42** | 81.82 | **85.76** | 86.12 | **61.5** | **2.22** |
| LLaVA-NeXT-7B | Greedy | 29.2 | 7.91 | 158.13 | 88.69 | 88.51 | 64.3 | 2.14 |
| | **Ours** | **25.8** | **6.52** | 152.87 | **88.92** | **89.08** | **56.8** | **2.57** |
| Qwen2-VL-7B | Greedy | 20.8 | 7.58 | 87.41 | 89.27 | 88.91 | 29.7 | 3.75 |
| | **Ours** | **17.8** | **5.14** | 82.51 | **89.45** | **89.15** | **25.3** | **3.77** |
| Qwen3-VL-8B | Greedy | 51.6 | 9.84 | 316.58 | 89.19 | 88.82 | 35.4 | 4.28 |
| | **Ours** | **45.2** | **8.52** | 310.59 | **89.80** | **89.62** | **32.6** | **4.32** |
| InternVL3-8B | Greedy | 21.2 | 7.07 | 99.97 | 88.16 | 87.81 | 48.8 | 2.74 |
| | **Ours** | **16.8** | **4.94** | 104.04 | **88.31** | **87.95** | **47.6** | **2.76** |

Table 2. Performance evaluation of LLaVA-1.5-7B on the subset of MME benchmark, where **bold** and underlined indicate the best and second best results respectively.

| | Exist. ↑ | Count ↑ | Pos. ↑ | Color ↑ | Total ↑ |
|---|---|---|---|---|---|
| Greedy | 195.00 | 158.33 | 123.33 | 155.00 | 631.66 |
| Beam | 195.00 | 136.66 | 123.33 | 145.00 | 599.99 |
| VCD | 190.00 | 163.33 | 118.33 | 143.33 | 614.99 |
| CODE | 195.00 | 131.66 | 133.33 | 150.00 | 609.99 |
| AGLA | 190.00 | 153.33 | 118.33 | 160.00 | 621.66 |
| VAF | 185.00 | 151.66 | 133.33 | 155.00 | 624.99 |
| VAR | 185.00 | 143.33 | 143.33 | 165.00 | 636.66 |
| SPARC | 195.00 | 158.33 | 123.33 | 153.33 | 629.99 |
| ONLY | 190.00 | 150.00 | 121.66 | 165.00 | 626.66 |
| VHR | 195.00 | 158.33 | 123.33 | 153.33 | 629.99 |
| **Ours** | 195.00 | 153.33 | 133.33 | 160.00 | **641.66** |

only adjusts the attention weights of LVLMs, without modifying the model architecture or inference process. This design enables training-free and plug-and-play deployment in different LVLM series.

## 4. Experiments

### 4.1. Experimental Setup

**Models and Baselines.** We evaluate our method on the most recent series from five popular LVLMs: LLaVA-1.5-7B (Liu et al., 2024a), LLaVA-NeXT-7B (Liu et al., 2024b), Qwen2-VL-7B (Wang et al., 2024), Qwen3-VL-8B (Bai et al., 2025), and InternVL3-8B (Zhu et al., 2025). To demonstrate the effectiveness of our ILVAD method in mitigating hallucination, we compare it with two baseline decoding methods, four well-established decoding adjustment methods and four state-of-the-art attention intervention methods. (1) Baseline decoding method: (i) Greedy decoding selects the token with the highest probability at each step; (ii) Beam search decoding keeps the top-k most likely sequences at each step to find a high-quality output. (2) Decoding adjustment method: (i) VCD (Leng et al., 2024) contrasts the output distribution generated from the original and perturbed images; (ii) CODE (Kim et al., 2024) performs contrastive decoding using self-generated descriptions; (iii) AGLA (An et al., 2025) masks irrelevant areas and performs collaborative decoding combining the output of the original image and the masked image; (iv) ONLY (Wan et al., 2025) selectively enhances textual output through single-layer network intervention. (3) Attention intervention method: (i) VAF (Yin et al., 2025) enhances attention to all visual tokens while suppressing attention to system tokens; (ii) VAR (Kang et al., 2025) identifies the attention sink and proportionally redistributes its attention to other visual tokens; (iii) SPARC (Jung et al., 2025) uses attention differences in time steps to identify and reinforce key visual tokens; (iv) VHR (He et al., 2025) uses the vision-aware head divergence to extract visually sensitive attention heads for enhancement.

Table 3. Performance evaluation on LLaVA-Bench (In-the-Wild) benchmark, scored by GPT-4o via pairwise response comparison, where **bold** indicate the best results.

| Model | Accuracy↑ | Detailedness↑ | Naturalness↑ |
|---|---|---|---|
| LLaVA-1.5-7B | 5.450 | 5.833 | **7.017** |
| **w/ ILVAD** | **5.833** | **6.133** | 6.867 |
| LLaVA-NeXT-7B | 5.400 | 6.417 | **7.767** |
| **w/ ILVAD** | **5.467** | **6.633** | 7.633 |
| Qwen2-VL-7B | 5.233 | **5.992** | 6.110 |
| **w/ ILVAD** | **5.558** | 5.940 | **6.197** |

Table 4. Ablation study on visual evidence and text enhancement, where **bold** indicate the best results.

| Setting | CHAIR$_S$↓ | CHAIR$_I$↓ | POPE-A↑ | POPE-F1↑ |
|---|---|---|---|---|
| Baseline | 48.6 | 13.52 | 84.50 | 85.22 |
| w/o Visual | 49.4 | 13.44 | 84.44 | 85.12 |
| w/o Text | 34.8 | 9.73 | 85.59 | 85.85 |
| **Ours** | **32.6** | **9.42** | **85.76** | **86.12** |

**Benchmarks.** We conduct extensive experiments on five benchmarks: (1) **CHAIR** (Rohrbach et al., 2018) evaluates object hallucination through image captioning, where LVLMs are required to generate captions for 500 randomly selected images from the validation split of the MSCOCO (Lin et al., 2014) dataset, and hallucinated objects in the generated captions are measured; (2) **POPE** (Li et al., 2023) is a benchmark for assessing object hallucination in LVLMs, which measures perceptual accuracy through yes-or-no questions about the presence of specific objects in images; (3)**MMHal-Bench** (Sun et al., 2024) evaluates multimodal hallucination by measuring factual inconsistency between model responses and visual content using human-annotated image–question pairs with fine-grained scores; (4) **MME** (Fu et al., 2023) is a comprehensive benchmark for LVLMs that evaluates object-level hallucinations through *existence* and *count*, and attribute-level hallucinations through *position* and *color*; (5) **LLaVA-Bench** (Liu et al., 2024a) contains 24 images covering complex scenes, memes, and sketches, together with 60 challenging questions designed to evaluate multimodal understanding. More details are provided in Appendix A.

**Implementation Details.** We set $\tau = 5$ for all experimental settings in our experiments. The enhancement strength $\alpha$ is set to 3 for LLaVA-NeXT-7B, and 5 for LLaVA-1.5-7B, Qwen2VL-7B, Qwen3VL-8b and InternVL3-8B. We set $\beta = 1$ for all discriminative benchmarks such as POPE and MME, while setting $\beta \in [0.2, 0.5]$ for descriptive benchmarks such as CHAIR, which targets hallucination suppression in long sequences. To balance efficiency with hallucination mitigation performance, we select the first 10 tokens generated for each sample to extract the visual evidence saliency map. To ensure a fair comparison, the results of

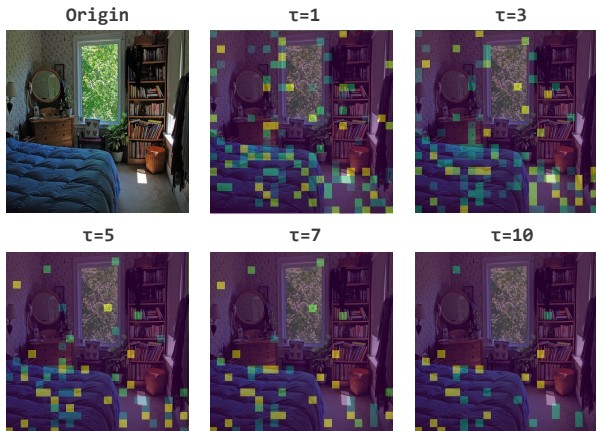

Figure 3. Visualization on the impact of $\tau$ towards visual evidence saliency map.

Table 5. Impact evaluation on $\tau$, where **bold** and underlined indicate the best and second best results respectively.

| Setting | CHAIR$_S$ ↓ | CHAIR$_I$ ↓ | Len. | POPE-A ↑ | POPE-F1 ↑ |
|---|---|---|---|---|---|
| $\tau = 1$ | 44.6 | 10.25 | 105.54 | 84.40 | 85.31 |
| $\tau = 3$ | **25.3** | **6.65** | 75.17 | 85.25 | 85.97 |
| $\tau = 5$ | 32.6 | 9.42 | 81.82 | 85.76 | 86.12 |
| $\tau = 7$ | 33.4 | 9.79 | 86.77 | 85.55 | 85.78 |
| $\tau = 10$ | 38.8 | 11.45 | 89.75 | **86.09** | **86.28** |

all methods are reported under the condition that the base model, the prompts, and the generation parameters remain consistent. The comparison methods all follow the default parameters provided in their original papers. To reduce randomness, we use greedy decoding for all the comparison methods. We further provide more implementation details in Appendix B.

### 4.2. Experimental Results

**Hallucination Benchmarks.** Table 1 reports the performance of our method on three hallucination benchmarks, including CHAIR, POPE, and MMHal-Bench. Compared with other hallucination mitigation methods, our method achieves the best performance on 5/6 hallucination evaluation metrics and ranks second on the remaining one. Specifically, our method only underperforms the AGLA method on the POPE F1 Score, but outperforms all baselines on every other metric. Compared with the baseline on LLaVA-1.5-7B, i.e., greedy decoding, our method reduces the CHAIR score by 31.6% and improves POPE and MMHal-Bench by 1.27% and 9.12%, respectively. Moreover, our method is training-free and plug-and-play. It remains effective in mitigating hallucinations across diverse model architectures, even on state-of-the-art models, e.g., Qwen3-VL-8B and InternVL3-8B. In Appendix C.1, more experimental results on other advanced LVLMs further demonstrate the effectiveness of our method in mitigating hallucinations.

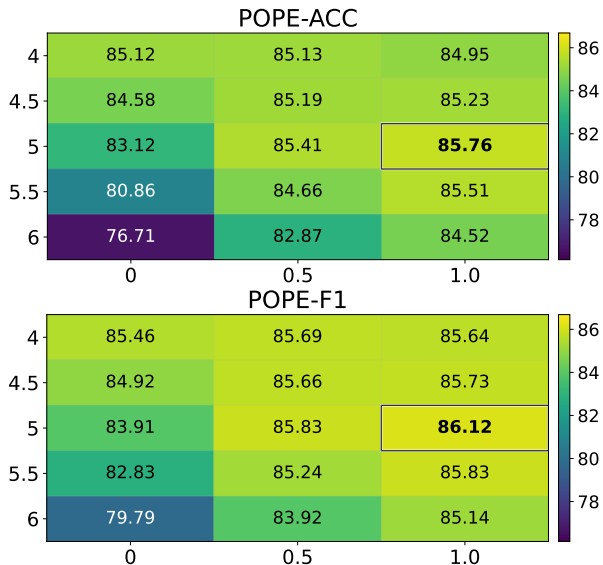

POPE-ACC / POPE-F1 heatmaps

Figure 4. Impact evaluation on $\alpha$ and $\beta$, with the black boxes highlighting the best results. The vertical axis represents $\alpha$, and the horizontal axis represents $\beta$.

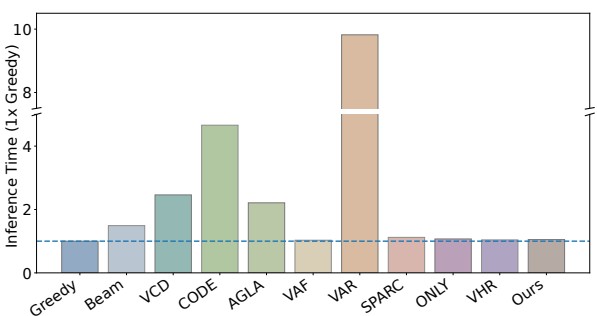

Figure 5. Comparison of inference times for different methods.

**Comprehensive Benchmarks.** We evaluate our method on a subset of the MME benchmark using LLaVA-1.5-7B. MME includes both object-level and attribute-level subtasks, providing a more detailed evaluation. As shown in Table 2, our method achieves the highest performance in the Exist subtask and is competitive with the strongest baseline in all remaining metrics. Although our method does not rank first in each subtask, it achieved the highest total score among all methods. This indicates that our approach consistently enhances multimodal capabilities across different evaluation dimensions, rather than being optimized for a single category. In order to further evaluate the quality of responses generated by our method, we conduct evaluations on LLaVA-Bench across three models. As shown in Table 3, our method effectively improves the accuracy of responses, while performing comparably to or even better than the baseline model in terms of detailedness and naturalness. This indicates that our method improves performance on more challenging tasks while maintaining response quality. In

Appendix C.2, the experimental results on long-form generation quality further indicate that our method improves the faithfulness and informativeness of responses while mitigating hallucinations.

### 4.3. Further Analysis

**Ablation Study.** In Table 4, we conduct ablation studies on LLaVA-1.5-7B to investigate the effectiveness of evidence-guided visual and text enhancement. We evaluate the performance of enhancing only visual evidence tokens and only text tokens on the CHAIR and POPE benchmarks. Results show that enhancing only text tokens cannot improve the model's understanding of visual information due to the lack of explicit visual evidence. Enhancing only visual evidence tokens effectively mitigates hallucinations, and further incorporating evidence-guided text enhancement leads to additional improvements in hallucination mitigation. The results confirm that our method effectively enhances visual evidence and selectively emphasizes text tokens that are strongly grounded in visual evidence.

**Impact of Hyperparameters.** We explore the impact of hyperparameters $\tau$, $\alpha$ and $\beta$ on LLaVA-1.5-7B through experimental evaluations and visualizations. Specifically, $\tau$ controls the filtering of significant tokens in each layer to influence the quality of the visual evidence saliency map. Figure 3 shows the visualization on the impact of $\tau$ towards visual evidence saliency map. Briefly, as the threshold $\tau$ increases, fewer visual tokens are selected, promoting the model to focus its attention on the most salient visual evidence. We further evaluate the performance of our method under different setting of $\tau$ in Table 5. Although a larger $\tau$ effectively improves performance on POPE, the gain on CHAIR is relatively modest. We thus select $\tau = 5$ to strike a balance between performance across different benchmarks. In addition, we also evaluate the performance of our method under different setting of $\alpha$ and $\beta$ in Figure 4. Our method is relatively stable for $\alpha$, and we just fix $\alpha = 5$ for most models. When beta is set to 0, our method suppresses attention to all text, which harms normal output. In experiments, it is necessary to choose the optimal parameter $\beta$. We further analyze the impact of the first-token window $T$ through experimental evaluations in Appendix C.3.

**Inference Time Analysis.** Our method only intervenes in the attention during the inference stage, so it incurs almost no additional inference overhead. Figure 5 shows the runtime comparison between our method and all baseline methods on LLaVA-1.5-7B. Our method has nearly the same inference overhead as the baseline model, demonstrating the efficiency of our approach.

**Case Study.** Qualitative analysis further demonstrates the effectiveness of ILVAD in mitigating hallucinations. We present some specific examples in Appendix E.2.

# 5. Conclusion

In this work, we found that LVLMs tend to hallucinate when they pay insufficient attention to the correct visual evidence and gradually forget it during the generation process. To address this issue, we conducted an in-depth study of the attention patterns of LVLMs toward visual tokens at different layers and proposed a hallucination mitigation method via inter-layer visual attention discrepancy. Our approach helps the model identify the correct visual evidence and enhance it. Additionally, we augment text that is faithful to visual evidence, further suppressing incorrect language priors. Our method is training-free and plug-and-play. Multiple benchmark evaluations conducted on five recently released models show that our method can consistently mitigate hallucinations in different LVLMs over various architectures.

# Acknowledgements

This work was supported by the National Natural Science Foundation of China under Grants U24A20322, 62576094, and 62576214, also partly supported by the Big Data Computing Center of Southeast University, the Guangdong Basic and Applied Basic Research Foundation under Grant 2024B1515020109.

# Impact Statement

This paper proposed ILVAD to promote research on mitigating hallucinations in Large Vision-Language Models. There are many potential societal consequences of our work, none of which we feel must be specifically highlighted here.

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

# A. Benchmarks and Metrics

To comprehensively evaluate the effectiveness of our method, we conduct extensive experiments on the following benchmarks and metrics.

**CHAIR.** The Caption Hallucination Assessment with Image Relevance (CHAIR) metric (Rohrbach et al., 2018) evaluates object hallucinations in LVLMs through image captioning. Given an input image $I$, the model is prompted to generate a free-form caption describing the visual content. The benchmark uses 500 randomly selected images from the MSCOCO (Lin et al., 2014) validation set, and object hallucinations are identified by comparing the objects mentioned in the generated captions with the ground-truth object annotations. CHAIR reports two metrics: the sentence-level hallucination rate ($\text{CHAIR}_S$) and the object-level hallucination rate ($\text{CHAIR}_I$). Specifically, the two metrics are defined as follows:

$$\text{CHAIR}_S = \frac{|\{\text{captions w/ hallucinated objects}\}|}{|\{\text{all captions}\}|},$$
$$\text{CHAIR}_I = \frac{|\{\text{hallucinated objects}\}|}{|\{\text{all mentioned objects}\}|}. \tag{11}$$

**POPE.** POPE (Li et al., 2023) is a widely used benchmark for assessing object hallucinations in LVLMs using a binary visual question answering (VQA) formulation. For each image $I$, the model is asked a set of yes-or-no questions regarding the presence of specific objects, such as "Is there a <object> in the image?". POPE constructs three subsets—*random*, *popular*, and *adversarial*—based on different strategies for generating negative samples. Each image is associated with 6 questions, resulting in a total of 3,000 test instances. Model performance is evaluated using standard classification metrics, including accuracy, precision, recall, and F1 score, computed over the binary predictions.

**MMHal-Bench.** MMHal-Bench (Sun et al., 2024) is a benchmark designed to evaluate multimodal hallucination with an emphasis on fine-grained factual consistency between model responses and visual content. Given an image–question pair $(I, Q)$, the model generates a free-form textual response $A$. The response is then evaluated using GPT-4o as an automatic judge, which assigns graded hallucination scores based on the degree of factual inconsistency between the generated content and the visual evidence.

**MME.** MME (Fu et al., 2023) is a comprehensive benchmark for evaluating LVLMs across object-level and attribute-level hallucinations using instruction-based yes-or-no questions. It consists of four subsets: *existence* and *count* for object-level hallucinations, and *position* and *color* for attribute-level hallucinations. Each subset contains 30 images, with two questions per image, resulting in 60 questions per subset. For each image–question pair $(I, Q)$, the model outputs a binary answer. Following the official protocol, performance is measured using both accuracy and accuracy+, where accuracy is computed over individual questions, and accuracy+ measures the proportion of images for which both associated questions are answered correctly. The final score combines these two metrics to reflect both local and image-level consistency.

**LLaVA-Bench.** LLaVA-Bench (In-the-Wild) (Liu et al., 2024a) evaluates multimodal understanding and dialogue capability in LVLMs through open-ended visual question answering. The benchmark consists of 24 images spanning diverse and challenging scenarios, including complex scenes, memes, and sketches, together with 60 carefully designed questions. Due to the open-ended nature of the task, model outputs are evaluated using GPT-4o as an automatic evaluator, which assesses the responses in terms of accuracy, detailedness, and naturalness.

**CapMAS.** CapMAS (Lee et al., 2025) is a recently proposed benchmark for evaluating hallucinations in hyper-detailed image captioning. It proposed Factuality and Coverage as fine-grained evaluation metrics to assess the correctness and completeness of generated descriptions, respectively. Factuality measures the degree of truthfulness and the absence of hallucinated content, and Coverage evaluates the comprehensiveness of the visual content in the caption.

**CLAIR and ALOHa.** CLAIR (Chan et al., 2023) measures the overall quality by evaluating the consistency between the generated caption and the reference caption. It directly uses LLMs to generate scores, rating based on the likelihood that the generated caption and the reference caption describe the same image. ALOHa (Petryk et al., 2024) measures the hallucinations in captioning models. It uses LLMs for open-vocabulary object extraction and utilizes semantic embeddings for fine-grained similarity matching.

## B. Implementation Details

In evaluation on CHAIR, we sampled three different sets of images using different random seeds. Each set includes 500 images from the MSCOCO validation set, with the prompt "Please describe this image in detail." for LVLMs to generate captions. In evaluation on POPE and MME, we do not constrain the model to generate only one single token. Instead, the model typically generates a complete answer with an explanation. Therefore, we first construct the saliency map by the initial 10 tokens, then discard the original response, and generate the final answer under the guidance of the saliency map. In evaluation on LLaVA-Bench, MMHal-Bench and CapMAS, we use GPT-4o as the evaluator. The prompt templates are shown in Appendix D. All open-source LVLMs that we used in the experiments come from the official checkpoints on the Hugging Face Model Hub.

## C. More Experimental Results and Analyses

### C.1. Evaluation on Other Advanced LVLMs

In Table 6, we report the evaluation of CHAIR and POPE benchmarks on other advanced LVLMs, including Qwen2.5-VL-7B[1], Qwen3.5-9B[2] and GLM-4.1V-9B[3], which are released from January 2025 to February 2026. Our method effectively mitigates hallucinations in all model architectures. This indicates that ILVAD generalizes well to newer and larger models.

Table 6. Performance evaluation across other advanced LVLMs on hallucination benchmarks, where **bold** indicate the best results.

| Model | CHAIR$_S$ ↓ | CHAIR$_I$ ↓ | Len. | POPE-A ↑ | POPE-F1 ↑ |
|---|---|---|---|---|---|
| Qwen2.5-VL-7B | 24.6 | 7.13 | 144.48 | 83.34 | 80.26 |
| **w/ ILVAD** | **21.6** | **6.79** | 137.09 | **83.91** | **81.14** |
| Qwen3.5-9B | 54.1 | 10.55 | 326.47 | 89.79 | 89.47 |
| **w/ ILVAD** | **50.6** | **9.49** | 317.98 | **89.95** | **89.69** |
| GLM-4.1V-9B | 26.2 | 6.36 | 145.73 | 89.18 | 88.68 |
| **w/ ILVAD** | **17.6** | **4.18** | 136.48 | **89.38** | **88.90** |

### C.2. Evaluation on CapMAS Benchmark

To further evaluate the performance of our method in mitigating hallucinations in long-form generation for LVLMs, we perform the experiments on the CapMAS (Lee et al., 2025) benchmark. Specifically, we used the IIW-400 (Garg et al., 2024) dataset, which contains 400 images paired with highly detailed, hallucination-free captions. Expect the metrics proposed by CapMAS, i.e., Factuality and Coverage, we further adopt the CLAIR (Chan et al., 2023) and ALOHa (Petryk et al., 2024) metrics to achieve a more comprehensive evaluation of hallucinations in long-form generation. The final scores are evaluated using GPT-4o. More details of the benchmark and metrics are provided in Appendix A.

Table 7. Performance evaluation of LLaVA-1.5-7B on CapMAS benchmark, scored by GPT-4o, where **bold** indicates the best results.

| Method | ALOHa | Factuality | Coverage | CLAIR | Avg. |
|---|---|---|---|---|---|
| Greedy | 43.76 | 62.78 | 33.30 | 50.70 | 47.64 |
| VHR | 44.05 | 62.49 | 25.80 | **54.80** | 46.79 |
| Ours | **46.78** | **65.45** | **34.80** | 51.90 | **49.73** |

Table 7 presents the results on ALOHa, Factuality, Coverage, CLAIR, and the average over these four metrics. We use LLaVA-1.5-7B as the baseline model and further compare our method with VHR, which achieved the second best performance on CHAIR in Table 1. The results show that our method outperforms the baselines on most metrics, and underperforms only on CLAIR compared with VHR. Overall, our method improves the average score by over 4.39% over the baseline. This indicates that our method performs effectively on more recent benchmarks that are better suited for evaluating long-form generation quality. In particular, the improvements on Factuality and Coverage indicate that our method improves the faithfulness and informativeness of long-form generation.

---

[1]Qwen2.5-VL-7B: `https://huggingface.co/Qwen/Qwen2.5-VL-7B-Instruct`
[2]Qwen3.5-9B: `https://huggingface.co/Qwen/Qwen3.5-9B`
[3]GLM-4.1V-9B: `https://huggingface.co/zai-org/GLM-4.1V-9B-Base`

## C.3. Impact of the First-token Window $T$

We further analyze the impact of first-token window $T$ on LLaVA-1.5-7B through experimental evaluations in Table 8. When $T = 1, 5$, ILVAD performs better on POPE and CHAIR, but the average length of CHAIR captions is abnormally short. This indicates that a smaller $T$ extracts insufficient visual information, leading to content loss in long text generation. In contrast, $T = 15, 20$ stabilizes the length, but increases the risks of hallucination. The results suggest that our choice of $T = 10$ is reasonable.

Table 8. Impact evaluation on $T$, where **bold** and underlined indicate the best and second best results respectively.

| Setting | CHAIR$_S \downarrow$ | CHAIR$_I \downarrow$ | Len. | POPE-A $\uparrow$ | POPE-F1 $\uparrow$ |
|---|---|---|---|---|---|
| $T = 1$ | **23.5** | 7.65 | 67.89 | **86.24** | **86.18** |
| $T = 5$ | 25.8 | **7.49** | 67.51 | 85.60 | 85.82 |
| $T = 10$ | 32.6 | 9.42 | 81.82 | 85.76 | 86.12 |
| $T = 15$ | 34.4 | 9.93 | 84.93 | 85.72 | 86.10 |
| $T = 20$ | 36.2 | 10.01 | 85.54 | 85.74 | 86.06 |

# D. Details on the GPT-4o Evaluation

**LLaVA-Bench Evaluation.** To evaluate the performance of LVLMs on LLaVA-Bench (In-the-Wild), we use GPT-4o as the evaluator. The prompt template adapted from (Gong et al., 2024) is shown in Table 9, with an additional metric, *Naturalness*, introduced to assess the fluency and coherence of the generated language. For each sample, GPT-4o is provided with the original image, the baseline LVLM output, and the output generated by ILVAD. The evaluation considers three key aspects: accuracy, detailedness, and naturalness, with particular emphasis on assessing the reduction of hallucinations in the ILVAD-enhanced responses relative to the baseline.

**MMHal-Bench Evaluation.** For each sample, we provide GPT-4o with the image contents extracted from OpenImages annotations, the question, a standard human-generated answer, and the response to be evaluated. Consistent with the original paper (Sun et al., 2024), the prompt[4] defines hallucination and provides representative examples, instructing GPT-4o to analyze and rate the response.

**CapMAS Evaluation.** Following the official CapMAS (Lee et al., 2025) evaluation protocol, we collect model responses from five different prompts: "Describe the given image in a very detailed manner", "Provide a detailed description of the specified image", "Elaborate on the details of the image provided", "Offer an in-depth description of the given image", and "Thoroughly describe the features of the specified image". Then we use Meta-LLaMA-3-8B-Instruct[5] to merge them into a single caption for evaluation. The prompt used for caption merging is shown in Table 10. We also provide the prompts used for GPT-4o evaluation of ALOHa, Factuality, Coverage, and CLAIR metics in Table 11.

# E. More Qualitative Results

## E.1. Visualization of Visual Evidence Saliency Map

In Figure 6, we provide more examples of visual evidence saliency maps. Our method produces more focused and interpretable saliency maps that highlight visual evidence relevant to the query text.

## E.2. Case Study.

We extend qualitative evaluation across three architectures to demonstrate the effectiveness of ILVAD method in mitigating hallucinations. Figures 7-9 present various qualitative examples between greedy decoding and ILVAD for LLAVA-1.5-7B, LLaVA-NeXT-7B and Qwen2-VL-7B. Through the integration of ILVAD, the captions generated by LVLMs exhibit improved fidelity to visual content while maintaining richness and informativeness.

---

[4]For the complete prompt, please refer to: https://github.com/llava-rlhf/LLaVA-RLHF
[5]Meta-LLaMA-3-8B-Instruct: https://huggingface.co/meta-llama/Meta-Llama-3-8B-Instruct

Table 9. Prompt used for GPT-4o evaluation on LLaVA-Bench.

| **Prompt** |
| --- |
| You are required to score the performance of two AI assistants in describing a given image. You should pay extra attention to the hallucination, which refers to the part of descriptions that are inconsistent with the image content, such as claiming the existence of something not present in the image or describing incorrectly in terms of the counts, positions, or colors of objects in the image. 

 Please rate the responses of the assistants on a scale of 1 to 10, where a higher score indicates better performance, according to the following criteria: 
 1: Accuracy: whether the response is accurate with respect to the image content. Responses with fewer hallucinations should be given higher scores. 
 2: Detailedness: whether the response is rich in necessary details. Note that hallucinated descriptions should not count as necessary details. 
 3: Naturalness: assess the language quality, focusing on: fluency of sentence structure, appropriateness of word choice, smoothness of language flow, absence of awkward or unnatural phrasing. 

 Please output the scores for each criterion, containing only two values indicating the scores for Assistant 1 and 2, respectively. The two scores are separated by a space. Following the scores, please provide an explanation of your evaluation, avoiding any potential bias and ensuring that the order in which the responses were presented does not affect your judgment. 
 [Question] 
 {} 
 [End of Question] 
 [Assistant 1] 
 {} 
 [End of Assistant 1] 
 [Assistant 2] 
 {} 
 [End of Assistant 2] 

 Output format: 
 Accuracy: 
 Reason: 
 Detailedness: 
 Reason: 
 Naturalness: 
 Reason: |

Table 10. Prompt used for caption merging with Meta-LLaMA-3-8B-Instruct.

| **Prompt** |
| --- |
| **System:** 
 This is a hard problem. Carefully summarize in ONE detailed caption based on the following 5 captions by different people describing the same image. Be sure to describe everything, and avoid hallucination. Provide the detailed caption in the format {Detailed caption}. 
 **User:** 
 Caption 1: {generated_captions[0]} 
 Caption 2: {generated_captions[1]} 
 Caption 3: {generated_captions[2]} 
 Caption 4: {generated_captions[3]} 
 Caption 5: {generated_captions[4]} |

Table 11. Prompts used for GPT-4o evaluation of ALOHa, Factuality, Coverage, and CLAIR.

| Metric | Prompt |
|---|---|
| **ALOHa** | You are an assistant that parses visually present objects from an image caption. Given an image caption, you list ALL the objects visually present in the image or photo described by the captions. Strictly abide by the following rules:
- Include all attributes and adjectives that describe the object, if present
- Do not repeat objects- Do not include objects that are mentioned but have no visual presence in the image, such as light, sound, or emotions
- If the caption is uncertain about an object, YOU MUST include '(possibly)' after the object
- If the caption thinks an object can be one of several things, include 'or' and all the possible objects
- Always give the singular form of the object, even if the caption uses the plural form |
| **Factuality** | **Proposition Decomposition (System):** I want to verify if the given CAPTION is accurate. To assist with this verification, decompose the given CAPTION into atomic propositions. All parts of the caption must be broken down into propositions. The outputs should follow the format: '1. proposition one 2. proposition two 3. proposition three'
**Visual Verification (System):** Your role is to determine whether the given propositions are True or False based on the provided image and its description. The outputs should follow the format: '1. True/False 2. True/False 3. True/False 4. True/False 5. True/False ...'
**User:** Description: {caption} Propositions: {questions} **Image:** {image} |
| **Coverage** | **System:** Your role is to answer the given questions based on the provided caption. I want to measure the amount of information in the caption. Therefore, if the correct answer to the question cannot be determined from the caption, you should answer it with "I don't know". Do not use your own knowledge in your response. Do not use information that can be inferred from the question itself. Only use the information provided in the caption. Answer the question by directly selecting the letter of the corresponding option. Do not repeat the question.
**User:** Caption: {caption}. Questions: {questions} |
| **CLAIR** | You are trying to tell if a candidate set of captions is describing the same image as a reference set of captions.
Candidate set: {candidate captions}. Reference set: {reference captions}
On a precise scale from 0 to 100, how likely is it that the candidate set is describing the same image as the reference set? (JSON format, with a key "score", value between 0 and 100, and a key "reason" with a string value.) |

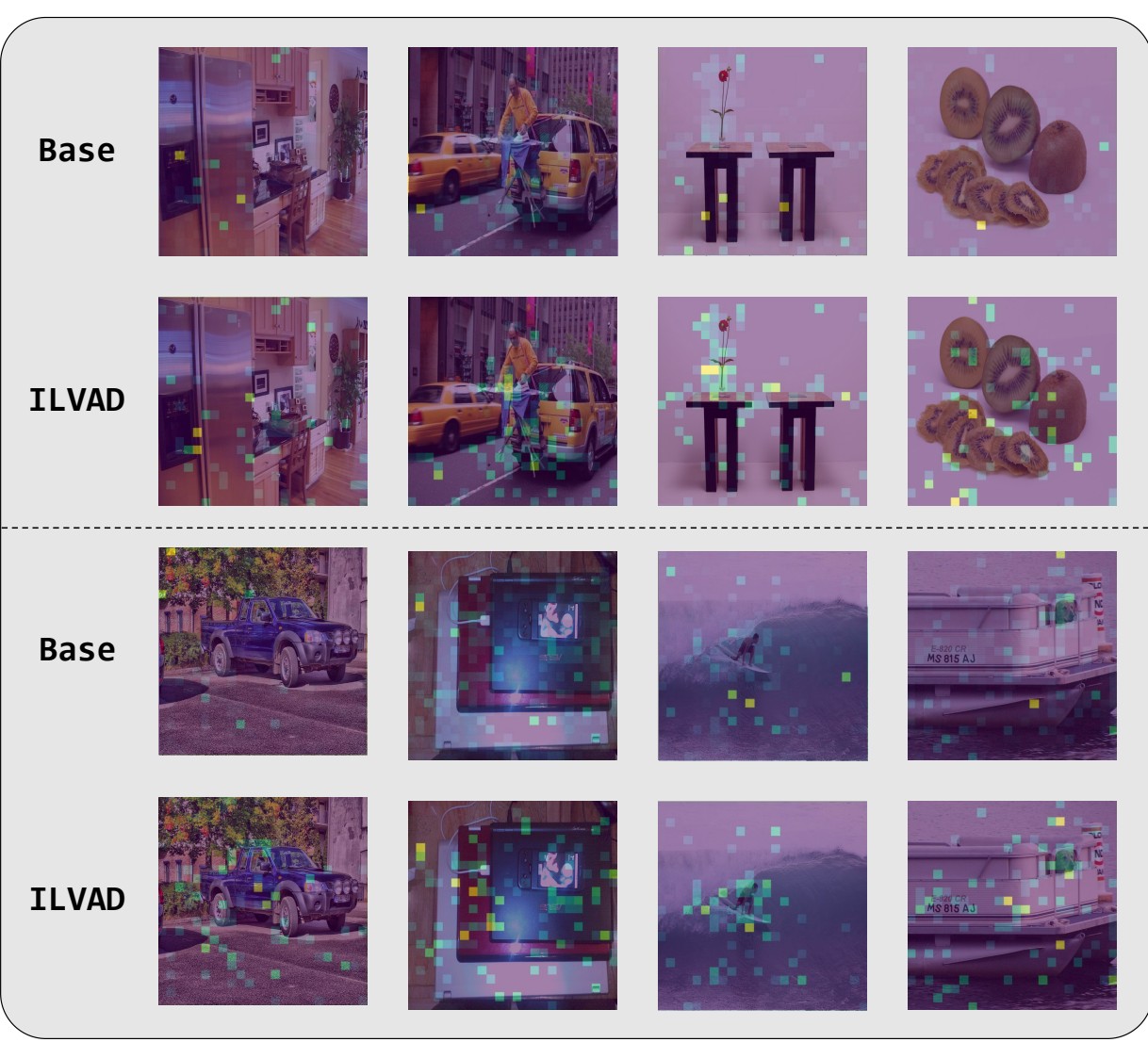

Figure 6. Visualization of visual evidence saliency maps. The queries of images above are "Is there a cup in the image?", "Describe this photo in detail.", "Which wooden stool has a vase with red flower on it?", and "Is the kiwi in the front fresh or dried?", respectively. All the queries of images below are "Please describe this image in detail.". Our method yields more interpretable saliency maps by emphasizing visually grounded evidence relevant to the query.

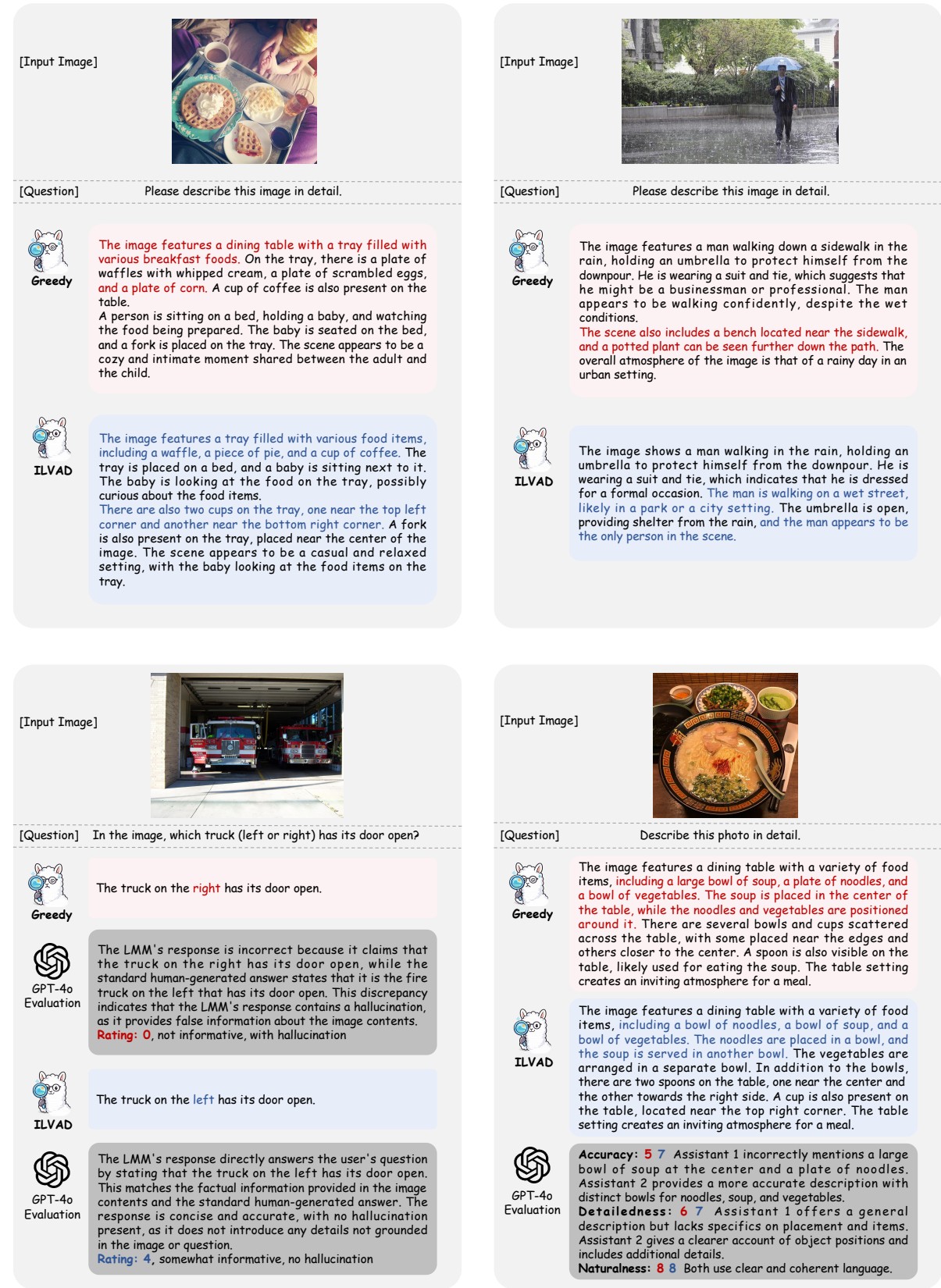

Figure 7. Case study for LLAVA-1.5-7B. The hallucinated text generated by the baseline (Greedy) and the corresponding real text generated by our method (ILVAD) are highlighted in red and blue fonts, respectively.

[Input Image]

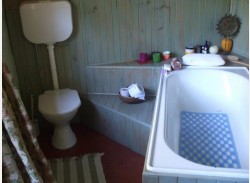

[Question]    Please describe this image in detail.

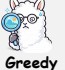
**Greedy**

The image captures a quaint, small bathroom bathed in a soothing palette of blue and white. Dominating the left side of the frame is a pristine white toilet, its lid closed, standing in stark contrast to the vibrant blue of the floor. Adjacent to the toilet, a white sink with a blue checkered pattern on its side adds a touch of whimsy to the room.
...
The walls of the bathroom are painted in a calming shade of blue, complementing the blue checkered pattern on the floor and bathtub. The floor is covered in a blue and white checkered rug, echoing the pattern on the bathtub and adding a sense of continuity to the room. The rug's placement under the sink and bathtub suggests careful planning to ensure comfort and functionality.

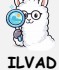
**ILVAD**

The image captures a quaint, small bathroom bathed in a soothing blue-green color scheme. Dominating the scene is a pristine white toilet, standing on the left side of the room. Its white tank is mounted on the wall, blending seamlessly with the overall color scheme. Adjacent to the toilet, a white bathtub with a blue checkered mat rests on the right side of the room.
...
The shelf, with its rustic charm, adds a touch of warmth to the cool blue-green room. The floor beneath is covered in a rug, its pattern adding a dash of color and texture to the space. The rug's placement under the bathtub suggests a thoughtful design, ensuring comfort and safety.

[Input Image]

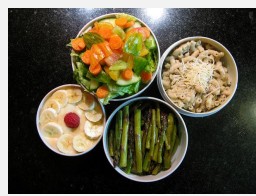

[Question]    Please describe this image in detail.

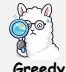
**Greedy**

The image captures a delightful array of four bowls... Starting from the top left, there's a bowl brimming with a fresh salad. The salad is a vibrant mix of orange carrots, green lettuce, and red radishes, all tossed in a light dressing.
Moving to the top right, we have a bowl filled with pasta...
In the bottom left, there's a bowl filled with a creamy soup. The soup is a rich blend of bananas, raspberries, and green beans, creating a colorful and appetizing display.
Finally, in the bottom right, we have a bowl filled with asparagus...

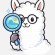
**ILVAD**

The image captures a delightful array of four bowls... In the top left, there's a bowl brimming with a vibrant salad, its contents a mix of fresh carrots and lettuce, their colors contrasting beautifully against the black background.
Adjacent to it, in the top right, is a bowl filled with pasta...
Moving to the bottom left, there's a bowl filled with sliced bananas. The bananas are arranged in a circular pattern, their bright yellow color standing out against the black countertop.
Finally, in the bottom right, there's a bowl filled with asparagus...

[Input Image]

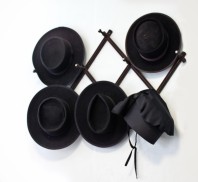

[Question]    How many black hats are there in the image?

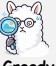
**Greedy**

There are four black hats in the image.

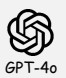
**GPT-4o Evaluation**

The LMM's response claims there are four black hats in the image, while the standard human-generated answer states there are five black hats. This discrepancy indicates that the LMM's response contains a hallucination. The response is somewhat informative, but the incorrect count of hats undermines its reliability.
**Rating: 1**, somewhat informative, with hallucination

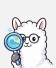
**ILVAD**

There are five black hats in the image.

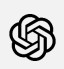
**GPT-4o Evaluation**

The LMM's response accurately states that there are five black hats in the image, which aligns with the standard human-generated answer. The response is concise and directly answers the question without any false claims or additional information that could lead to hallucination. Therefore, the response is both informative and free of hallucination.
**Rating: 5**, very informative, no hallucination

[Input Image]

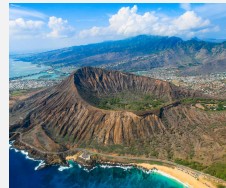

[Question]    What is the name of this famous sight in the photo?

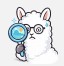
**Greedy**

The image you've provided appears to show a view of the Na Pali Coast on the island of Kauai in Hawaii. The Na Pali Coast is a famous sight known for its dramatic cliffs, lush vegetation, and rugged coastline. It is a popular destination for hiking, kayaking, and sightseeing.

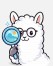
**ILVAD**

The image you've provided appears to show a view of the Diamond Head volcanic crater, also known as Leahi, located in Honolulu, Hawaii. It's a famous landmark and a popular tourist destination known for its distinctive shape and the panoramic views it offers from the top.

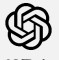
**GPT-4o Evaluation**

**Accuracy: 3 10** Assistant 1 incorrectly identified the location as the Na Pali Coast. The image actually depicts Diamond Head, a volcanic crater in Honolulu, Hawaii, which was correctly identified by Assistant 2.
**Detailedness: 5 8** Assistant 1 provides some general details about the location, such as its dramatic cliffs and popularity for hiking and sightseeing, but these are not specific to what is actually shown in the image. Assistant 2 gives more relevant and location-specific information, mentioning the volcanic crater, its alternative name "Leahi" location in Honolulu, and its panoramic views, which are factual and match the image.
**Naturalness: 8 8** Reason: Both responses use fluent and natural language with appropriate sentence structure and word choice.

Figure 8. Case study for LLaVA-NeXT-7B. The hallucinated text generated by the baseline (Greedy) and the corresponding real text generated by our method (ILVAD) are highlighted in red and blue fonts, respectively.

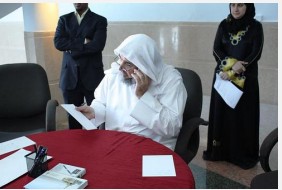

[Input Image]

[Question]    Please describe this image in detail.

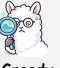

**Greedy**

The image depicts a formal setting with three individuals. The central figure is an elderly man wearing a white robe and a white head covering, sitting at a round table covered with a red tablecloth. He is holding a piece of paper and appears to be engaged in a conversation on a mobile phone. To his left, there is a man dressed in a dark suit and tie, standing with his arms crossed. He is wearing glasses and has a serious expression on his face. To his right, there is a woman wearing a black dress with a yellow pattern on it. She is holding a white paper and appears to be observing the scene. The background features a beige wall and a few chairs around the table.

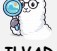

**ILVAD**

The image depicts a formal event or meeting taking place in a room with a red tablecloth. There are three people in the scene: a man in traditional white clothing, a woman in a black dress, and a man in a suit. The man in white is seated at a table, engaged in a conversation on a cell phone. The woman in black is standing to the right of the the man in white, while the man in the suit is standing in the background. The setting appears to be a professional or formal gathering, possibly in a business or governmental context. The room has a modern yet formal ambiance, with chairs and a table setup.

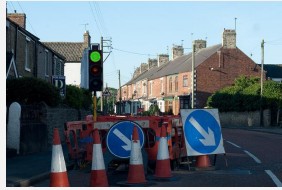

[Input Image]

[Question]    Please describe this image in detail.

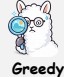

**Greedy**

The image depicts a street scene with a traffic light displaying a green light. There are several traffic cones and signs set up on the street, indicating some sort of roadwork or construction. The cones are arranged in a way that suggests they are directing traffic around the the work area. In the background, there are several houses with chimneys, and a few people can be seen walking on the the sidewalk. The houses appear to be residential, and the street is relatively quiet. The overall atmosphere of the scene suggests that the area is undergoing some form of improvement or or maintenance work.

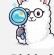

**ILVAD**

The image depicts a street scene in a small town with a row of houses and a traffic light. The traffic light is green, and there are several traffic cones and signs blocking the road. The street is lined with houses, and there is a for sale sign on one of the houses. The scene is set during the day, and the houses are of a traditional style.

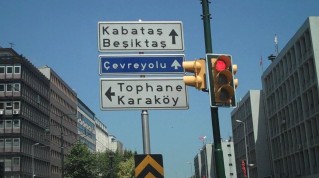

[Input Image]

[Question]  What colors are the traffic signs at the top, in the middle, and at the bottom?

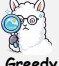

**Greedy**

The the top, middle, and bottom traffic signs are white with black text.

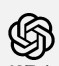

**GPT-4o Evaluation**

The LMM response states that all three traffic signs ("top, middle, bottom") are white with black text. According to the image contents and the standard human-generated answer, the one in the middle is blue. Therefore, the LMM incorrectly claims that the middle sign is white, which is a hallucination. The response is somewhat informative, but it contains a factual error .
**Rating: 1**, somewhat informative, with hallucination

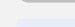

**ILVAD**

The the top sign is white with black lettering, the middle sign is blue with white lettering, and the bottom sign is white with black lettering.

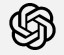

**GPT-4o Evaluation**

The LMM response accurately describes the colors of the traffic signs as observed in the image contents. The LMM adds that the white signs have black lettering and the blue sign has white lettering. This is a reasonable and likely accurate detail and it does not contradict the image contents. The response is informative and provides a bit more detail than the standard answer.
**Rating: 6**, very informative with good analysis or reasoning, no hallucination

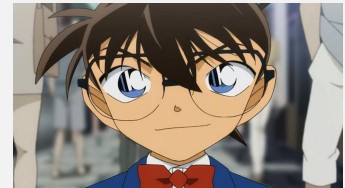

[Input Image]

[Question]    What is the name of the character in the image?

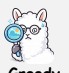

**Greedy**

The character in the image is Monkey D. Luffy, the protagonist of the anime and manga series "One Piece".

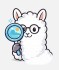

**ILVAD**

The character in the image is Conan Edogawa, a fictional anime and manga character. He is the protagonist of the "Detective Conan" series, which is also known as "Case Closed" in English.

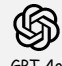

**GPT-4o Evaluation**

**Accuracy: 1 10** Assistant 1 incorrectly identified the character as "Monkey D. Luffy", which is a clear hallucination, as the character in the image is Conan Edogawa from "Detective Conan". Assistant 2 correctly identified the character without any hallucination.
**Detailedness: 1 9** Assistant 1 only provided the character's name and the series, but since the identification is wrong, these details are not valid. Assistant 2 not only named the character but also included context: his role as a protagonist and the alternative series name, \"Case Closed\".
**Naturalness: 8 9** Both responses exhibit good fluency and appropriate word choice with smooth sentence structure. Assistant 2's response flows slightly better due to the additional contextual information, which is naturally integrated and free of awkward phrasing.

Figure 9. Case study for Qwen2-VL-7B. The hallucinated text generated by the baseline (Greedy) and the corresponding real text generated by our method (ILVAD) are highlighted in red and blue fonts, respectively.

