# OpenReview forum: "Finding the Correct Visual Evidence Without Forgetting: Mitigating Hallucination in LVLMs via Inter-Layer Visual Attention Discrepancy"
_ICML.cc/2026/Conference — ICML 2026 regular_

### Official Review · Reviewer_hjYD · 2026-02-28

**Soundness:** 4
**Presentation:** 3
**Significance:** 2
**Originality:** 3
**Overall Recommendation:** 5
**Confidence:** 4

**Summary:**

The paper analyzes the relationship between attention maps and hallucinations in LVLMs and finds that hallucinations often occur when the model allocates insufficient attention to relevant visual evidence. Additionally, LVLMs do not consistently attend to visual evidence across layers, showing strong inter-layer discrepancy. The authors propose a train-free method that uses this discrepancy to enhance this attention and show that the method effectively mitigates hallucinations and improves visual understanding in LVLMs.

**Compliance With Llm Reviewing Policy:**

Affirmed.

**Final Justification:**

The rebuttal addressed my concerns

**Key Questions For Authors:**

See the weaknesses

**Limitations:**

I couldn't find a section that clearly discusses the limitations of the paper, and I encourage the authors to add such a section.

**Strengths And Weaknesses:**

__Strengths__:

The paper presents a clear analysis that motivates an edit in the weight space to reduce hallucinations. The paper is well-written, and the results show strong benefits of using this method for various applications. The ablations are very useful for understanding the robustness of this method.

__Weaknesses__:

1. The analysis does not discuss other impacts of the presented edit. How does this approach change the behaviour of existing benchmarks (e.g., common benchmarks that can be found in VLMEvalKit). If this approach harms the results in these cases, it makes it less useful for the general case, while if this approach helps/doesn't change the result, this approach can be a very significant contribution to the community.

2. The analysis is done on relatively old and small models. With Qwen2.5, Qwen3, Qwen3.5, and GLM models, it is not clear how useful this approach is in a more modern setting. While I still think that this is a useful and interesting direction even for older models, I would like to know how relevant this approach is for newer and larger models. An analysis of the relations between the approach and scale/model version can make it significantly stronger.

3. Missing references: Some relevant papers need to be mentioned and compared (e.g., [1,2]). Some of these methods can be combined with the presented approach, and I am curious to see whether they can work together to improve the results further.


[1] Huang et al., OPERA: Alleviating Hallucination in Multi-Modal Large Language Models via Over-Trust Penalty and Retrospection-Allocation, CVPR 2024

[2] Jiang et al., Interpreting and Editing Vision-Language Representations to Mitigate Hallucinations, ICLR 2025

---

> ### Author Rebuttal · Authors · 2026-03-31
>
> **Thanks for your time and effort in handling this paper. We respond to your questions point by point.**
>
> ## Weakness 1: More common benchmarks.
>
> **Table R6: Comparison on Common Benchmarks from VLMEvalKit**
> | |MM-Vet|VizWiz|HallusionBench|
> |-|-|-|-|
> |LLaVA-1.5-7B|26.7|50.2|48.5|
> |+Ours|**29.4**|**50.4**|**48.8**|
>
> In **Table R6**, we report three common benchmarks from VLMEvalKit, including MM-Vet, VizWiz and HallusionBench [1-3], which cover diverse capabilities such as recognition, OCR, knowledge, language generation, spatial awareness, and mathematical computation. **Our method achieves better performance than the baseline on these benchmarks**. In particular, our method achieves a **10.1%** improvement on MM-Vet. This preliminarily proves that our method does not harm general benchmark performance. We will conduct more general benchmark evaluations on more model architectures in the final version of the paper.
>
> [1] Yu et al., MM-Vet: Evaluating Large Multimodal Models for Integrated Capabilities, ICML 2024
>
> [2] Gurari et al., VizWiz-Priv: A Dataset for Recognizing the Presence and Purpose of Private Visual Information in Images Taken by Blind People, CVPR 2019
>
> [3] Guan et al., HallusionBench: An Advanced Diagnostic Suite for Entangled Language Hallucination & Visual Illusion in Large Vision-Language Models, CVPR 2024
>
> ## Weakness 2: Evaluations on newer and larger models.
>
> **Table R7: Comparison on Newer and Larger Models**
> |Model|CHAIRs$\downarrow$|CHAIRi$\downarrow$|Len.|POPE-ACC|POPE-F1|
> |-|-|-|-|-|-|
> |Qwen2.5-VL-7B-Instruct|24.6|7.13|144.48|83.34|80.26|
> |+Ours|**21.6**|**6.79**|137.09|**83.91**|**81.14**|
> |Qwen3-VL-8B-Instruct|51.6|9.84|316.58|88.16|87.81|
> |+Ours|**45.2**|**8.52**|310.59|**88.31**|**87.95**|
> |Qwen3.5-9B|54.1|10.55|326.47|89.79|89.47|
> |+Ours|**50.6**|**9.49**|317.98|**89.95**|**89.69**|
> |GLM-4.1V-9B-Base|26.2|6.36|145.73|89.18|88.68|
> |+Ours|**17.6**|**4.18**|136.48|**89.38**|**88.90**|
>
> In **Table R7**, we report the evaluation of CHAIR and POPE metrics on **four newer models released from January 2025 to February 2026. Our method effectively mitigates hallucinations across all model architectures.** This preliminarily proves that our method generalizes well to newer and larger models (8B, 9B).
>
> ## Weakness 3: Missing references.
>
> We appreciate the reviewer for pointing out these relevant works. **Both methods are representative approaches in this area, and we will add the corresponding discussion and citations in the final version of the paper**.
>
> **Table R8: Comparison with OPERA and PROJECTAWAY**
> |Model|CHAIRs$\downarrow$|CHAIRi$\downarrow$|Len.|POPE-ACC|POPE-F1|
> |-|-|-|-|-|-|
> |OPERA|45.0|12.64|83.15|85.62|85.96|
> |PROJECTAWAY|41.6|12.09|86.48|84.62|85.31|
> |Ours|32.6|9.42|81.82|85.76|86.12|
> |Ours+OPERA|38.4|10.98|78.05|**85.92**|86.00|
> |Ours+PROJECTAWAY|**30.2**|**7.73**|104.63|85.78|**86.19**|
>
> Specifically, OPERA mitigates hallucinations by introducing an over-trust penalty on the model logits during beam-search decoding, together with a rollback strategy. PROJECTAWAY reduces hallucinations by linearly orthogonalizing image features with respect to hallucinated object features. In **Table R8**, we report both direct comparisons and combinations with our method.
>
> **The results show that our method outperforms both methods on all reported metrics, which further demonstrates its effectiveness in mitigating hallucinations.** When combined with OPERA, it achieves an improvement on POPE-ACC. When combined with PROJECTAWAY, several metrics are further improved. However, during CHAIR evaluation, we observe that combining with PROJECTAWAY tends to generate repetitive outputs (Len.=104.63), which negatively affects generation quality. We suggest that the image editing process may remove useful fine-grained details, making it harder for the model to obtain sufficient and reliable visual evidence.
>
> Although the combination with PROJECTAWAY is not yet ideal, the improvement in POPE metrics still suggests **meaningful complementarity between different types of hallucination mitigation methods**. We will provide a more detailed analysis and discussion in the final version of the paper.
>
> ## Limitation 1: Section that clearly discusses the limitations.
>
> Thanks for the suggestion. In the paper, the current evaluation mainly focuses on object level hallucinations, which may be insufficient for assessing the long-form generation quality. We will consider more fine-grained evaluations of factual consistency and content completeness. **We will add the above discussions of the limitation in the final version of the paper.**

---

> > ### Author Rebuttal · Reviewer_hjYD · 2026-04-03
> >
> > The authors resolved my concerns with strong results. I believe that they can be added to the paper to improve it, and willing to raise my score.

---

> > > ### Author Response · Authors · 2026-04-03
> > >
> > > Dear Reviewer **hjYD**,
> > >
> > > We are happy that your concerns are fully resolved. Thanks again for your valuable comments and suggestions. We will make the suggested modifications in the final version.
> > >
> > > Regards from the authors.

---

### Official Review · Reviewer_J9Eo · 2026-03-11

**Soundness:** 3
**Presentation:** 3
**Significance:** 3
**Originality:** 3
**Overall Recommendation:** 4
**Confidence:** 4

**Summary:**

This paper addresses the issue of hallucinations in Large Vision-Language Models (LVLMs), where generated responses are inconsistent with visual content. The authors identify that hallucinations often occur when models pay insufficient attention to correct visual evidence and gradually "forget" it during the generation process. To exploit this, the authors propose a training-free, plug-and-play method that constructs a saliency map to identify and enhance critical visual tokens while emphasizing text tokens strongly grounded in visual evidence. Experimental results across five benchmarks and various architectures, demonstrate that ILVAD consistently mitigates hallucinations and improves multimodal understanding with minimal inference overhead.

**Compliance With Llm Reviewing Policy:**

Affirmed.

**Final Justification:**

The rebuttal addressed my concerns.

**Key Questions For Authors:**

See the weakness.

**Limitations:**

Hyperparameter Sensitivity across Architectures: The effectiveness of ILVAD appears dependent on manual tuning of key parameters. For instance, the enhancement strength must be varied between models (e.g., 3 for LLaVA-NeXT-7B vs. 5 for Qwen2-VL-7B). The authors should discuss the lack of a universal parameter setting and the resulting tuning cost for new models.

**Strengths And Weaknesses:**

**Strength:**
1. The paper is well-written and easy to follow, providing a clear motivation and a logically structured methodology.
2. The authors evaluated their method across five diverse benchmarks and five LVLM architectures. The proposed ILVAD method demonstrates robust generalization and powerful hallucination mitigation.
3. The method is training-free and plug-and-play, requiring no architectural modifications. Crucially, it introduces minimal additional inference overhead, maintaining a runtime nearly identical to greedy decoding.

**Weakness:**
1. Potential Bias in Saliency Map: The reliance on a fixed, short window of early tokens assumes that the model's initial attention patterns are sufficient to identify all relevant visual evidence for the entire response. However, the authors does not address whether this static saliency map remains effective for long-form generation where the required visual evidence might shift beyond what is captured in the first 10 tokens.
2. For discriminative benchmarks such as POPE and MME, the model's correctness is typically determined by the very first generated token (e.g., 'Yes' or 'No'). Could the authors clarify how the saliency map—which requires 10 tokens to be constructed—is used to mitigate hallucinations in these single-token response tasks?

---

> ### Author Rebuttal · Authors · 2026-03-31
>
> **Thanks for your time and effort in handling this paper. We respond to your questions point by point.**
>
> ## Weakness 1: Potential bias in saliency map.
>
> We appreciate the reviewer’s concern. We agree that the visual evidence required for long-form generation may change during generation. However, our method does not rely on the assumption that the first 10 generated tokens capture all relevant evidence for the entire response. Instead, it find and enhance visual evidence overlooked by the first 10 tokens via inter-layer discrepancy.
>
> Empirically, the saliency map extracted from the first 10 tokens is effective. We compare the average attention on each ground-truth evidence patch between the baseline map and our saliency map over the first 10 tokens. **Our saliency map increases the average attention from 0.036 to 0.071 (1.97×)**. This suggests that our method can already capture significant visual evidence at the early stage.
>
> **Table R3：The analysis of $T$**
> |T|CHAIRs$\downarrow$| CHAIRi$\downarrow$|Len.| Acc$\uparrow$|F1$\uparrow$|
> |-|-|-|-|-|-|
> |1|**23.5**|7.65|67.89*|**86.24**|**86.18**|
> |5|23.8|**7.49**|67.51*| 85.60 |85.82|
> |10|32.6|9.42|81.82| 85.76 |86.12|
> |15|34.4|9.93|84.93| 85.72 |86.10|
> |20|36.2|10.01|85.54| 85.74 |86.06|
>
>
> We also analyze $T$ in **Table R3**. When $T=1, 5$, POPE results are competitive, but the CHAIR descriptions are abnormally short (**67 v.s. 82**), indicating insufficient information for descriptive tasks. In contrast, $T=15, 20$ stabilizes length but increases hallucination risks. **The results suggest that our choice of $T=10$ is reasonable since it provides a good balance between generation quality and hallucination mitigation.**
>
> **Table R4: CHAIR metric for long-text generation model**
> |Model|CHAIRs$\downarrow$|CHAIRi$\downarrow$|Len.|
> |-|-|-|-|
> |Qwen3-VL-8B-Instruct|51.6|9.84|316.58|
> |+Ours|**45.2**|**8.52**|310.59|
> |Qwen3.5-9B|54.1|10.55|326.47|
> |+Ours|**50.6**|**9.49**|317.98|
>
> **Table R5: Performance on ALOHa, Factuality, Coverage and CLAIR**
> | Model|ALOHa|Factuality|Coverage|CLAIR|Avg.|
> |-|-|-|-|-|-|
> |LLaVA-1.5-7B|43.76|62.78|33.3|50.7|47.64|
> |VHR (ACL 2025)|44.05|62.49|25.8|**54.8**|46.79|
> |Ours|**46.78**|**65.45**|**34.8**|51.9|**49.73**|
>
> In addition, **Table R4** reports the results on CHAIR for long-text generation with the updated models. The average response length of Qwen3-VL-8B-Instruct reached 310.59 tokens, and Qwen3.5-9B reached 317.98 tokens. Our method improves the CHAIR metric by 12.91% and 8.26% on these two models, respectively. Moreover, in **Table R5**, our method remains effective on more recent long-form evaluation benchmarks, including ALOHa, Factuality, Coverage and CLAIR [1-3]. **These results suggest that, although the saliency map is constructed from an early fixed window, it remains practically effective for long-form generation.**
>
> [1] Lee et al., Toward Robust Hyper-Detailed Image Captioning: A Multiagent Approach and Dual Evaluation Metrics for Factuality and Coverage, ICML 2025
>
> [2] Petryk et al., ALOHa: A New Measure for Hallucination in Captioning Models, NAACL 2024
>
> [3]Chan et al., CLAIR: Evaluating Image Captions with Large Language Models, EMNLP 2023
>
> ## Weakness 2: Discriminative benchmarks.
>
> In our evaluation on POPE and MME, we do not constrain the model to generate only a single token. Instead, the model typically generates a complete answer with explanation. Thus, we first construct the saliency map by the initial 10 tokens, then discard the original response and generate the final answer under the guidance of the saliency map.
>
> Since our method does not modify the prefill stage, the prefill KV cache can be reused. Therefore, the additional inference overhead mainly comes from the first 10 decoding steps rather than extra first-token latency. **This issue is an implementation detail of the inference rather than a conceptual limitation of our method for discriminative benchmarks**. We will provide more detailed implementation details in the final version of the paper.
>
> ## Limitation 1: Hyperparameter sensitivity across architectures.
>
> Thanks for the comments. **We agree that cross-model hyperparameter selection is a practical and important issue**. Although ILVAD involves several hyperparameters, most of them were fixed in our experiments across different architectures: $\tau$=5, $T$=10, $\beta$=0.5 for descriptive tasks and $\beta$=1 for discriminative tasks. In practice, only $\alpha$ requires minor adjustment within range {3, 4, 5}. Therefore, for a new architecture under the same task setting, the tuning cost is limited to three trials.
>
> **We also provide analyses of these hyperparameters in Figure 3, Figure 4, Table 5, and Table R3, which offer preliminary evidence that our parameter choices are reasonable**. We will include additional analysis of hyperparameter settings and tuning costs on more model families in the final version.

---

> > ### Author Rebuttal · Reviewer_J9Eo · 2026-04-05
> >
> > The authors successfully addressed my concerns, so I keep my positive score.

---

> > > ### Author Response · Authors · 2026-04-05
> > >
> > > Dear Reviewer J9Eo,
> > >
> > > **We are happy that your concerns are fully resolved**. Thanks again for your valuable comments and suggestions. We will make the suggested modifications in the final version.
> > >
> > > Regards from the authors.

---

### Official Review · Reviewer_ux3F · 2026-03-12

**Soundness:** 3
**Presentation:** 3
**Significance:** 2
**Originality:** 3
**Overall Recommendation:** 4
**Confidence:** 4

**Summary:**

This paper addresses the issue of hallucinations in LVLMs caused by insufficient focus on correct visual evidence and the gradual "forgetting" of such evidence during the generation process. The proposed method extracts saliency maps by analyzing attention weights from early-generated tokens to visual tokens across layers, identifying consistently activated "visual evidence" tokens. These maps are then used to boost visual attention during generation to mitigate visual forgetting.

**Compliance With Llm Reviewing Policy:**

Affirmed.

**Final Justification:**

The rebuttal addressed my concerns.

**Key Questions For Authors:**

Please see the weaknesses.

**Limitations:**

Yes.

**Strengths And Weaknesses:**

### Strength
* The approach leverages saliency maps to bolster visual attention during the decoding stage, which effectively prevents critical visual information from being "diluted" or "forgotten" as the sequence length increases.
* The proposed method is training-free and introduces negligible computational overhead, ensuring it does not significantly impact inference latency.

### Weakness
* ILVAD requires extracting the attention weights from the first $T$ generated tokens to construct the saliency map. Consequently, if the model fails to attend to the correct visual regions during the initial generation phase (e.g., the first 10 tokens), the effectiveness of the subsequent enhancement may be significantly constrained.
* The process of identifying "visual evidence" requires filtering out "Attention Sinks." However, if the filtering mechanism is too aggressive, it may inadvertently suppress subtle but essential visual cues, leading to the loss of fine-grained information.
* The underlying concept—masking or suppressing non-salient image regions to mitigate abnormal attention—shares significant similarities with existing work [M]. The authors should more clearly articulate the unique technical contributions and distinctions of their method compared to this baseline.

[M] Mitigating Object Hallucinations in Large Vision-Language Models with Assembly of Global and Local Attention

---

> ### Author Rebuttal · Authors · 2026-03-31
>
> **Thanks for your time and effort in handling this paper. We respond to your questions point by point.**
>
> ## Weakness 1: Effectiveness when the model fails to attend to the correct visual regions.
>
> Although ILVAD constructs the saliency map from the first $T$ generated tokens, **it does not need the model to attend to all correct visual regions at early stage**. Instead, it is motivated by two empirical insights.
>
> **Table R2: Comparison on Baseline Map & Saliency Map**
> | |Avg. Attention|Weighted Recall|Weighted F1|
> |-|-|-|-|
> |Baseline Map|0.036|0.048|0.076|
> |Saliency Map|**0.071**|**0.136**|**0.177**|
>
> First, prior work [1] shows the early activation patterns of LVLMs, and we observe similar patterns in visual evidence. In **Table R2**, the baseline map over the first 10 generated tokens assigns an **Avg. Attention** of 0.036 to ground-truth evidence patches. This value is about **21** times higher than the uniform attention (1/576=0.0017). This suggests that some visual evidence is already attended within the first 10 tokens. However, the baseline map still yields low **Weighted Recall and F1**, indicating incomplete coverage of useful evidence.
>
> Second, this neglect is not uniform across layers. Although overall attention of baseline is insufficient, some layers still attend to important evidence. In **Figure 1**, for the sample involving the visual evidence “baseball”, the average attention over the first 10 tokens hardly covers the evidence, while certain layers assign substantially higher attention. In Figure 1 (iii), after filtering out attention sinks with abnormally large relative attention, visual evidence shows larger inter-layer deviation (**0.22**) than non-evidence tokens (**0.07**). **The saliency map built from this discrepancy improves Avg. Attention, Weighted Recall, and Weighted F1 by 1.97×, 2.83× and 2.33× over the baseline in Table R2, respectively. These results suggest that ILVAD can indeed identify more accurate visual evidence even when the early overall attention is insufficient.**
>
> In addition, we also analyze $T$ in **Table R3 of Reviewer J9Eo**. When $T=1, 5$, POPE results are competitive, but the CHAIR descriptions are abnormally short (**67 vs. 82**), indicating insufficient information. In contrast, $T=15, 20$ stabilizes length but increases hallucination risks. **The results support that our choice of $T=10$ is reasonable.**
>
> ## Weakness 2: Loss of fine-grained information.
>
> **To balance information filtering and preservation of fine-grained cues, we introduce the threshold $\tau$.**
>
> We have provided both visualization and quantitative analysis for different $\tau$ in **Figure 3** and **Table 5**. As $\tau$ increases, the amount of extracted visual information decreases. Smaller $\tau$ preserves more information but also introduces more distracting noise, resulting in worse performance on the discriminative benchmark POPE. Larger $\tau$ makes the retained information more concentrated on salient evidence, which improves POPE but harms performance on the descriptive benchmark CHAIR.
>
> In contract, our chosen setting $\tau=5$ performs well on both benchmarks. Moreover, the saliency map extracted with $\tau=5$ outperforms the baseline map across all three metrics in **Table R2. These results support that $\tau=5$ provides a reasonable balance between filtering irrelevant information and preserving fine-grained visual evidence**.
>
> ## Weakness 3: Unique contribution & distinction from AGLA.
>
> **We have discussed AGLA in the related work section and included it in our experimental comparisons**. Here we further clarify our contribution and distinction.
>
> **Unique contribution.** As shown in Figure 1, we analyze the relationship between visual evidence and hallucinations in LVLMs. Hallucinations often occur when the model allocates insufficient attention to relevant visual evidence, and this issue worsens as the model gradually forgets the evidence during generation. **Our core contribution is to identify and enhance overlooked visual evidence through inter-layer discrepancy, as discussed above.**
>
> **Distinction**. The two methods differ in both technical route and mechanism. First, our method is an **attention intervention** approach, whereas AGLA is a **decoding based** method. Second, **ILVAD effectively identify and enhance overlooked visual evidence through inter-layer discrepancy, while AGLA does not explicitly enhance the model’s internal attention to visual information.** AGLA relies on an external GradCAM plugin and text queries to mask non-salient tokens. When task-specific textual queries are unavailable, e.g. CHAIR, AGLA may remove useful fine-grained information and fail to guide the model toward overlooked visual evidence. This limitation is also reflected in **Table 1**, where AGLA performs worse on CHAIR (**46.4 vs. 32.6**).
>
> [1] Li et al., The Hidden Life of Tokens: Reducing Hallucination of Large Vision-Language Models via Visual Information Steering, ICML 2025

---

> > ### Author Rebuttal · Reviewer_ux3F · 2026-04-03
> >
> > Lack a comparison with MemVR [M], which similarly addresses the issue of visual evidence decay.
> >
> > [M] Zou et al, Look Twice Before You Answer: Memory-Space Visual Retracing for Hallucination Mitigation in Multimodal Large Language Models, ICML 2025.
> >
> >
> > Thanks to the authors for responding to my questions. My concerns are resolved, and I will update my recommendation.

---

> > > ### Author Response · Authors · 2026-04-05
> > >
> > > We appreciate the reviewer for pointing out this related work. Regarding this newly raised issue, we provide the following clarification and comparison.
> > >
> > > **MemVR is a representative approach in this area, and we will add the corresponding discussion and citation in the final version of the paper**. Specifically, MemVR method reinjects visual information into the Feed Forward Network (FFN) when the model exhibits high uncertainty during inference. These visual tokens serve as key-value memory to alleviate the “amnesia” about visual information. **However, this method only traces back the visual information that has already been captured at the middle trigger layer, but cannot help the model identify and enhance the important visual evidence that has been overlooked**.
> > >
> > > **Table R9: Comparison of the attention on visual evidence.**
> > > |Method|Avg. Attention|Weighted Recall|Weighted F1|
> > > |-|-|-|-|
> > > |MemVR|0.039|0.051|0.079|
> > > |Ours|**0.071**|**0.136**|**0.177**|
> > >
> > > As shown in **Table R9**, we compare the average attention to visual evidence and the Weighted Recall and F1 for the two methods. **Compared with MemVR, our method allocates more attention to visual evidence and achieves better Weighted Recall and F1**. We also record the model's average attention to visual evidence at each generation step. The MemVR method decreases from 0.067 to 0.023 (65.7%), while our method decreases from 0.099 to 0.051 (48.5%). **These statistics indicate that MemVR only provides more visual information but does not improve the model's attention to visual evidence**.
> > >
> > > **Table R10: Comparison with MemVR on CHAIR and POPE.**
> > > |Method|CHAIRs$\downarrow$|CHAIRi$\downarrow$|Len.|POPE-ACC|POPE-F1|
> > > |-|-|-|-|-|-|
> > > |MemVR|46.8|12.35|86.92|84.67|85.36|
> > > |Ours|**32.6**|**9.42**|81.82|**85.76**|**86.12**|
> > >
> > > **Table R11: Comparison with MemVR on MME.**
> > > |Method|Total|Exist.|Count|Pos.|Color|
> > > |-|-|-|-|-|-|
> > > |MemVR|634.99|195.00|158.33|128.33|153.33|
> > > |Ours|**641.66**|195.00|153.33|133.33|160.00|
> > >
> > > In **Table R10** and **Table R11**, we report the comparison of our method and MemVR on the CHAIR, POPE and MME benchmarks. **Our method outperforms MemVR on all three metrics, indicating that our method can more effectively mitigate hallucinations in diverse scenarios**.
> > >
> > > We hope this discussion addresses the reviewer’s new concern. If there are any further questions, we would be happy to provide detailed responses.

---

### Official Review · Reviewer_9JZe · 2026-03-12

**Soundness:** 3
**Presentation:** 3
**Significance:** 3
**Originality:** 3
**Overall Recommendation:** 4
**Confidence:** 5

**Summary:**

Unlike prior works that derive saliency maps from differences across decoding steps or across attention heads, this paper focuses on attention differences across layers to obtain saliency maps, and uses them during decoding to mitigate hallucinations. Specifically, it first selects attention heads and layers that are sensitive to visual tokens, and then amplifies visual tokens that receive high attention weights. To reduce the influence of sink tokens, it uses the changes in attention weights between adjacent layers as the final scores for constructing the saliency map. Experiments on CHAIR, POPE, MMHAL-Bench, as well as general benchmarks, show that the proposed method outperforms existing approaches.

**Compliance With Llm Reviewing Policy:**

Affirmed.

**Key Questions For Authors:**

1. In Equation (6), how does the method handle the case where a token is genuinely important and therefore highly activated across all layers, rather than being a spurious or hallucinatory signal?
2. CHAIR is not an appropriate benchmark for evaluating hallucinations in long-form generation for LVLMs. Does the proposed method still outperform baselines on more recent or more suitable evaluation benchmarks?

**Limitations:**

No.

**Strengths And Weaknesses:**

Strengths
1. Because MLLMs are being used for many high‑stakes decision‑making tasks, research on reducing their hallucinations is timely.
2. The method does not require training and can be applied easily.

Weaknesses
1. In Equation (6), how does the method handle the case where a token is genuinely important and therefore highly activated across all layers, rather than being a spurious signal?
2. CHAIR is not an appropriate benchmark for evaluating hallucinations in long-form generation for LVLMs. Does the proposed method still outperform baselines on more recent or more suitable evaluation benchmarks? [1,2,3]

[1] Lee et al., "Toward Robust Hyper-Detailed Image Captioning: A Multiagent Approach and Dual Evaluation Metrics for Factuality and Coverage" ICML 2025
[2] Petryk et al., "ALOHa: A New Measure for Hallucination in Captioning Models" NAACL 2024
[3] Chan et al., "CLAIR: Evaluating Image Captions with Large Language Models" EMNLP 2023

---

> ### Author Rebuttal · Authors · 2026-03-31
>
> **Thanks for your time and effort in handling this paper. We respond to your questions point by point.**
>
> ## Weakness 1 and Question 1: How to handle genuinely important and highly activated tokens.
>
> We appreciate the reviewer’s concern. Equation (6) is not intended to identify tokens that are highly activated across all layers. Instead, it is designed to find visual evidence that is overlooked by the model and enhance it. In our method, significantly activated tokens are defined as tokens whose attention in each layer exceeds a threshold in Equation (5). If a token is genuinely important and highly activated across all layers, it indicates that the model has already assigned sufficient attention. In such cases, no additional enhancement is needed.
>
> Moreover, to avoid disrupting the model’s existing focus on such already well attended tokens, we introduce the coefficient $\alpha$ to control the enhancement strength. The coefficient $\alpha$ serves as a trade-off parameter between enhancing visual evidence that is overlooked and preserving the model’s original attention distribution. Experiments on multiple hallucination benchmarks (**Table 1**) and comprehensive benchmarks (**Table 2, 3 and R1**) demonstrate the effectiveness of our strategy of identifying and enhancing overlooked visual evidence. The analysis of $\alpha$ in **Figure 4** further demonstrates that our choice of enhancement strength is reasonable.
>
> ## Weakness 2 and Question 2: More recent evaluation benchmarks.
>
> Thanks for the suggestion. We performed the experiments on the suggested benchmarks. Following the setup in [1], for each image in the IIW-400 dataset, we use different prompts to generate five captions and then employ Llama-3-8B to merge them into a single final caption for evaluation.
>
> **Table R1: Performance on ALOHa, Factuality, Coverage and CLAIR**
> | Model|ALOHa|Factuality|Coverage|CLAIR|Avg.|
> |--------------|------:|------:|-----------:|---------:|------:|
> |LLaVA-1.5-7B|43.76|62.78|33.3|50.7|47.64|
> |VHR| 44.05 |62.49|25.8|**54.8**|46.79|
> |Ours|**46.78**|**65.45**| **34.8**|51.9|**49.73**|
>
> **Table R1** presents the results on ALOHa, Factuality, Coverage, CLAIR, and the average over these four metrics. We use LLaVA-1.5-7B as the baseline model and further compare our method with VHR [2], which achieved the second best performance on CHAIR in **Table 1**. The results show that our method outperforms the baselines on most metrics, and underperforms only on CLAIR compared with VHR. Overall, our method improves the average score by over **4.39%** over the baseline.
>
> **These results indicate that our method performs effectively on more recent benchmarks that are better suited for evaluating long-form generation quality.** In particular, the improvements on **Factuality** and **Coverage** indicate that our method enhances both the **faithfulness and informativeness of long-form generation**. We will include more detailed experimental settings and analyses in the final version of the paper.
>
> [1] Lee et al., Toward Robust Hyper-Detailed Image Captioning: A Multiagent Approach and Dual Evaluation Metrics for Factuality and Coverage, ICML 2025
>
> [2] He et al., Cracking the code of hallucination in lvlms with vision-aware head divergence, ACL 2025

---

> > ### Author Rebuttal · Reviewer_9JZe · 2026-04-04
> >
> > The authors addressed all my concerns

---

> > > ### Author Response · Authors · 2026-04-04
> > >
> > > Dear Reviewer **9JZe**
> > >
> > > Thanks again for your time and effort in reviewing this paper. Thanks for your positive score. **We are happy that all of your concerns have been addressed**. Please consider raising your score. Thanks a lot for your support.
> > >
> > > Regards from the authors.

---

### Decision · Program_Chairs · 2026-04-30

**Decision:**

Accept (regular)

**Comment:**

After reviewing the paper, the reviewers’ comments, and the authors’ rebuttals, I recommend acceptance. The authors addressed all reviewer concerns thoroughly and professionally. They provided additional experiments on more recent benchmarks (ALOHa, Factuality, Coverage, CLAIR) and newer, larger models (Qwen3-VL, GLM-4.1V), demonstrating the method’s generalization and effectiveness. They clarified the handling of genuinely important tokens, the role of the early token window, and the distinction from related work (e.g., AGLA, MemVR). All reviewers acknowledged that their concerns were fully resolved, with several raising their scores accordingly. The paper is technically solid, training-free, and empirically well-supported, with no unresolved issues. Therefore, I recommend acceptance.